# NDR2 Kinase Regulates Microglial Metabolic Adaptation and Inflammatory Response: Critical Role in Glucose-Dependent Functional Plasticity

**DOI:** 10.3390/ijms262110630

**Published:** 2025-10-31

**Authors:** Beatriz Fazendeiro, Ivo Machado, Anabela Rolo, Paulo Rodrigues Santos, António Francisco Ambrósio, Paulo F. Santos, Hélène Léger

**Affiliations:** 1Coimbra Institute for Clinical and Biomedical Research (iCBR), Faculty of Medicine, University of Coimbra, 3000-548 Coimbra, Portugal; beatrizfazendeiro00@gmail.com (B.F.); afambrosio@fmed.uc.pt (A.F.A.); pfsantos@ci.uc.pt (P.F.S.); 2Center for Innovative Biomedicine and Biotechnology (CiBB), University of Coimbra, 3004-504 Coimbra, Portugal; 3Clinical Academic Centre of Coimbra (CACC), 3004-504 Coimbra, Portugal; 4Center for Neuroscience and Cell Biology (CNC-UC), University of Coimbra, 3004-504 Coimbra, Portugal; 5Doctoral Program in Experimental Biology and Biomedicine (PDBEB), Institute of Interdisciplinary Research, University of Coimbra, 3004-504 Coimbra, Portugal; 6Department of Life Sciences, University of Coimbra, 3030-790 Coimbra, Portugal; 7Laboratory of Immunology and Oncology, Center for Neuroscience and Cell Biology (CNC-UC), University of Coimbra, 3004-504 Coimbra, Portugal; 8Institute of Immunology, Faculty of Medicine (FMUC), University of Coimbra, 3004-504 Coimbra, Portugal; 9Center of Investigation in Environment, Genetics and Oncobiology (CIMAGO), Faculty of Medicine, University of Coimbra, 3001-301 Coimbra, Portugal; 10Association for Innovation and Biomedical Research on Light and Image (AIBILI), 3000-548 Coimbra, Portugal

**Keywords:** retina, diabetic retinopathy, microglia, inflammation, Nuclear Dbf2-Related (NDR) kinases, *Ndr2/Stk38l* gene, high-glucose, metabolism

## Abstract

Diabetic retinopathy (DR), a major complication of diabetes, is driven by chronic inflammation in which retinal microglial cells play a central role. The Hippo pathway kinases NDR1/2 regulate macrophage function, but their role in microglia and DR remain unknown. This study investigates the function of the NDR2 kinase in microglial cells under high-glucose (HG) conditions. Using CRISPR-Cas9, we partially knocked out the *Ndr2/Stk38l* gene in BV-2 mouse microglial cells and analyzed metabolic activity, phagocytosis, migration, and cytokine release. We confirmed NDR2 expression in microglia and observed increased levels under HG, suggesting a role in hyperglycemia-induced stress. *Ndr2/Stk38l* (*hereafter referred to as Ndr2*) downregulation impaired mitochondrial respiration and reduced metabolic flexibility, indicating defective stress adaptation. Functionally, microglia with a partial downregulation of *Ndr2* displayed reduced phagocytic and migratory capacity—both dependent on cytoskeletal dynamics. Moreover, *Ndr2* downregulation altered the secretory profile, elevating pro-inflammatory cytokines (IL-6, TNF, IL-17, IL-12p70) even under normal glucose levels. These findings identify NDR2 protein kinase as a key regulator of microglial metabolism and inflammatory behavior under diabetic conditions. By modulating immune and metabolic responses, NDR2 may contribute to the neuroinflammatory processes underlying DR. Targeting NDR2 function in microglia may offer novel therapeutic strategies to mitigate retinal inflammation and progression of DR.

## 1. Introduction

Diabetes and its associated retinal disease—diabetic retinopathy (DR)—are a rapidly increasing health, societal, and economic burden. About 93 million people worldwide have diabetic retinopathy [1], which affects 1 out of 3 persons with diabetes mellitus, making it the leading cause of vision loss among working-age adults.

DR is a complex disease with a chronic inflammatory component in the retina. Many genetic and environmental factors have been considered to contribute to its development [2].

Impaired glucose metabolism, a key feature of diabetes, leads to biochemical changes that cause retinal vascular damage, a hallmark of the disease, also inducing the activation of the retinal microglial cells and the production of reactive oxygen species (ROS) [3]. Microglial cells are mononuclear phagocytes like macrophages, the first line of defense of the nervous system and the resident immune cells of the central nervous system [4,5,6]. Activated microglia contribute to the progression of DR by releasing cytokines, chemokines, and other bioactive molecules that promote vascular leakage, abnormal blood vessel growth, and vascular and neural cell death in the retina [6]. Understanding the role of microglial cells and inflammation in DR has led to research into potential therapeutic strategies targeting these cells and the associated neuroinflammatory processes [6,7,8].

Approaches aimed at modulating microglial activation or reduce inflammation could potentially slow or prevent the progression of DR, complementing traditional approaches focused on blood glucose control and vascular health [9,10]. Several recent studies have highlighted the effect of molecules and bioactive compounds on the progression of DR, by reducing pro-inflammatory responses and promoting anti-inflammatory activity, primarily through microglial modulation [11,12].

The NDR (Nuclear Dbf2-Related) kinases encoded by *Ndr1/Stk38* and *Ndr2/Stk38l*, are serine/threonine protein kinases that are evolutionarily conserved and play essential roles in regulating various cellular processes such as cell growth and apoptosis [13,14,15]. Dysregulation of NDR kinase signaling has been implicated in various diseases, including cancer, neurodegenerative disorders, and metabolic diseases [16,17]. While the NDR kinases are not extensively studied in the context of inflammation compared to other kinases, there is emerging research linking NDR kinases to inflammation mediated by macrophages by regulating the pattern-recognition receptors and via their interactions with various signaling pathways such as the nuclear factor-κB [18,19,20,21,22,23].

In the retina, we have previously established that *Ndr2* and *Ndr1* deletion in mice leads to a concomitant increase in apoptosis and cell proliferation of terminally differentiated neurons [24] similar to the phenotype observed in the canine early retinal degeneration (erd) disease [25,26,27]. Enrichment analyses of the differentially expressed genes (DEGs) from (Ndr2) erd dog retinas present an early activation of immune response genes, such as genes encoding constituents of NLRP3 inflammasome, and common components of IL1-β, IL-18, and TLR4 inflammatory pathways [28]. In mouse retinas, enrichment analyses of the DEGs point out that *Ndr2* deletion causes deregulation of retinal stress- and inflammation-related gene expression with enrichment of genes associated with oxidative stress, and cytoskeleton misregulation [24]. New causal analysis of the DEGs highlighted the putative role of the NDR2 kinase in the inflammation pathways mTOR, CXCR4 and eIF17 [24,29]. In neurons, mitochondrial dysfunction, oxidative phosphorylation, PI3K/AKT, mTOR, eIF2, eIF4, and p70S6K signaling lead to an inflammatory response and are known to be regulated by the microglia [29]. However, the exact functions and mechanisms of NDR kinases in retinal inflammation and in the context of DR are not yet known.

This study aims to elucidate the role of NDR2 kinase as a new key player in microglial cells exposed to high glucose conditions. This will permit a better understanding of the regulation of microglial cells, and their surveillance and damage-sensing functions while exposed to metabolic stress, such as hypoglycemia in the pathology of DR [30,31].

## 2. Results

### 2.1. NDR Kinases Are Expressed by Microglial Cells

To demonstrate the expression and localization of native NDR kinases in microglial cells, we performed immunocytochemistry using commercially available antibodies targeting, respectively, the N-terminus (aa 1-100; NDR1/2 antibody (E-2) #sc-271703) and the C-terminus (aa 380-460; NDR2 antibody #STJ94368) of the human NDR2 kinase on human induced pluripotent stem cell (iPSC) -derived microglial cultures, BV-2 immortalized microglial cells, and mouse primary retinal microglial cultures [29]. In both cultures, the microglia phenotype was validated by immunocytochemistry using antibodies against markers of several cell types: (i) neuronal marker NeuN (negative staining), (ii) Müller cell marker vimentin (negative staining), (iii) astrocytes marker Glial Fibrillary Acidic Protein (GFAP; Figure 1A,B), and (iv) microglial cell marker ionized calcium-binding adapter molecule 1 (IBA1; Figure 1A,B).

We were able to stain both NDR kinases in human microglial cells (Figure 1C), while only the NDR2 antibody gave positive staining in mouse primary (Figure 1D) and immortalized microglial cells (Figure 1E). As shown in Figure 1C, NDR1 and NDR2 colocalized with the cytoskeletal protein IBA1 (ionized calcium binding adapter protein 1) in human iPSCs-derived microglial cells [32]. In contrast, NDR2 staining in mouse primary retinal microglia was predominantly localized at the cell periphery and at the tips of microglial processes (Figure 1D). NDR2 is localized in the cytoplasm in a peri-nuclear fashion in the immortalized microglial cell line BV-2. IBA1 is an actin-crosslinking protein, expressed specifically in microglial cells and macrophages, and it plays a crucial role in microglial cell migration, membrane ruffling, phagocytosis, and remodeling during immunological surveillance [32]. In summary, the data shown in Figure 1 indicate that both NDR kinases are expressed by microglial cells and that NDR2 is expressed in microglial cells in the cytoplasm in a peri-nuclear fashion, at the cell periphery and at the tips of microglial processes.

### 2.2. NDR2 Protein Expression Is Upregulated in Microglial Cells Exposed to HG

To determine the effect of high-glucose (HG) conditions on NDR2 protein levels, BV-2 cells were exposed to 30.5 mM of glucose for 7 h or two times 4 h periods of HG with a 4 h period in between exposed to control condition e.g normal glucose (CT e.g NG; 5.5 mM of glucose),hereafter referred to as the 12 h assay (Figure 2). Protein levels were then analyzed by Western blot. The results were normalized for calnexin, a loading control protein [31]. The exposure of BV-2 cells to HG conditions in the 7 h assay (CT: 24.0 ± 4.4 a.u.; HG: 83.0 ± 19.1 a.u.) and in the 12 h assay (CT: 26.1 ± 6.9 a.u.; HG: 64.2 ± 10.1 a.u.) caused a significant increase in NDR2 protein expression (Figure 3A).

To determine the impact of HG exposure on *Ndr2* mRNA levels in BV-2 cells, we conducted a qRT-PCR analysis. As shown in Figure 3B, no alterations were observed in the expression of *Ndr2* mRNA for the 7 h HG exposure (HG: 80.0 ± 0.1% of CT) when compared to CT conditions. However, a tendency towards an increase was found when cells were exposed to the 12 h assay (HG: 160.3 ± 34.0% of CT, *p* = 0.097). We also evaluated the impact of HG exposure on Ndr1 mRNA levels in BV-2 cells by qRT-PCR analysis and we found no alterations in the expression of Ndr1 for the 7 h HG (HG: 80.8 ± 23.1% of CT) and 12 h HG exposure (HG: 120.3 ± 51.0%) when compared to CT conditions (Appendix A). In summary, these results suggest that HG conditions upregulate the NDR2 protein level but not NDR1 protein level in microglial cells.

To specifically investigate the role of NDR2 kinase in microglial cell responses in the context of HG exposure, we conducted a CRISPR-Cas9 lipofectamine transfection of the all-in-one plasmid containing a sgRNA against the exon 7 of the *Ndr2* gene to disrupt the expression of NDR2 in BV-2 cells.

### 2.3. CRISPR-Cas 9 Induced a Downregulation of the Ndr2 Gene in BV-2 Cells

We transfected early passages BV-2 cells (p7) with the all-in-one plasmid VB230329-1589djw, targeting *Ndr2* exon 7 for 24 h followed by puromycin selection for 30 h. We designed sgRNA #7 to specifically target exon 7 of the *Ndr2* gene and evaluate its predicted performance using the CHOPCHOP design tool. The sgRNA exhibited an efficiency score of 46.27, coupled with limited off-target potential, showing only 2 and 1 predicted off-target transcripts with 1 and 3 mismatches, respectively [33,34]. Further in silico analysis using CRISPOR confirmed its specificity with a MIT specificity score of 52/100, exceeding the recommended minimum threshold for guide RNA specificity, and a high CFD score of 91/100, indicating a strong likelihood of on-target cleavage with minimal off-target activity [35,36,37,38]. Predicted off-target analysis, ran with the software CRISPOR v.5.2 (https://crispor.gi.ucsc.edu/ (accessed on 3 September 2025)) allowed us to identified the top 15 potential off-targets (*intron:Slco6d*, *intergenic:Zfp958-Shcbp1*, *intergenic:Rnf144b-4930471G24Rik*, *intergenic:Vip-Fbxo5*, *intergenic:Shank2-Gm14372*, *intergenic:Gng4-B3galnt2*, *intergenic:Trim52-1700128A07Rik*, *intergenic:Naaladl2-Nlgn1*, *intergenic:Cd70-Tnfsf14*, *intergenic:Grik2-Ascc3*, *intergenic:A530006G24Rik-Foxa2*, *intergenic:Gpr165-Pgr15l*, *exon:Stk32a*, *intergenic:D330037F02Rik-Gadl1*, *intergenic:Cul3-1700016L21Rik*) (Appendix A).

Following transfection of early passage BV-2 cells with the all-in-one plasmid VB230329-1589djw containing sgRNA #7, puromycin selection was performed to enrich successfully edited cells. After puromycin selection, we set up approximately 40 clonal cultures. Based on the phenotype of the clonal populations, we selected 10 of them for genomic analysis to check for insertion/deletion (Indels) after non-homologous end-joining (NHEJ) repair events using Sanger sequencing. We were able to classify the clones into three categories, each presenting an indel in the region near the PAM sequence (CCA in blue, Table 1). We chose to work with one clone of each of the distinct categories that were most likely to generate possible knockouts: clone 13, 19, and clone 22. This confirmed the presence of Indels consistent with non-homologous end joining repair mechanisms. Two validation assays were performed to confirm that, indeed, we had a disruption of the *Ndr2* gene. Firstly, we assessed the *Ndr2* mRNA levels by qRT-PCR in these clonal populations compared with wild type (WT) BV-2 cells. As we can see in Figure 4A, all clones presented a decrease in the mRNA levels when compared to the WT and control group (Clone 13: 50 ± 10% of CT; Clone 19: 23.5 ± 4.6% of CT; Clone 22: 26.5 ± 2.9% of CT) (see Table 1 for BV-2 *Ndr2* clones). Clones 19 and 22 showed a greater reduction in *Ndr2* mRNA levels compared to clone 13, which led to their selection for further analysis. We performed immunocytochemistry on clones 19 and 22 to evaluate the NDR2 protein expression under CT condition using the commercially available antibody targeting the C-terminus (aa 380-460; NDR2 antibody #STJ94368) of the human NDR2 kinase (Figure 4B). We also observed a decrease in the NDR2 protein levels in both selected clones when compared to WT BV-2 expression (Clone 19: 5.2% of WT; Clone 22: 40.9% of WT). Since we have amplification signals considering both qRT-PCR results and NDR2 expression from immunocytochemistry, we cannot consider that these clones present a real knockout of the *Ndr2* gene. However, there is a clear downregulation of the *Ndr2* gene for both clones.

### 2.4. High Glucose and Ndr2 Downregulation Effects on Cell Viability and Metabolism

To assess the effects of *Ndr2* downregulation on the viability of BV-2 cells, we conducted an Alamar blue assay, which relies on resazurin reduction to resorufin [39]. A non-statistically significant decrease in resazurin reduction was observed in both *Ndr2* downregulated BV-2 cells when compared to WT BV-2 cells (Clone 19 7 h CT: 77.6 ± 12.0% control; Clone 22 7 h CT: 68.9 ± 6.5% of control) (Figure 5A). When WT cells were exposed to HG, no statistically significant alterations were registered compared to the CT condition (WT 7 h HG: 83.8 ± 5.8% of WT CT). However, when *Ndr2* downregulated cells were exposed to 7 h incubation with HG, a significant decrease was detected when compared to WT cells in CT conditions (Clone 19 7 h HG: 67.9 ± 9.1% of WT CT; Clone 22 7 h HG 67.0 ± 5.8% of WT CT). In the case of the 12 h assay, only clone 19 exhibited a significant decrease in resazurin reduction when exposed to HG: WT 12 h HG: 95.3 ± 4.1% of WT CT; Clone 19 12 h CT: 57.6 ± 9.7% of WT CT, Clone 19 12 h HG: 48.8 ± 12.8% of WT CT; Clone 22 12 h CT: 77.1 ± 14.8% of WT CT; Clone 22 12 h HG 75.9 ± 18.5% of WT CT) (Figure 5B). Additionally, we incubated cells with 25 mM of mannitol (M) along with 5.5 mM of D-glucose already in cell culture medium, with mannitol being used as an osmotic control to ensure that the observed effects from HG exposure were not due to osmotic fluctuations [40]. Our results demonstrate that mannitol did not affect the viability of the cells (Appendix A).

As the Alamar Blue assay provided an indirect assessment of cell viability and metabolic activity, we sought to directly evaluate cell death in WT, clone 19, and clone 22 BV-2 cells under both CT and HG conditions. To this end, we performed flow cytometry analysis. We evaluated the FITC-annexin-V (AV) and 7-amino-actinomycin D (7-AAD) staining and grouped them into living cells (AV-/7-AAD-), early apoptotic cells (AV+/7-AAD-), late apoptotic cells (AV+/7-AAD+), and necrotic cells (AV-/7-AAD+), as shown in Appendix A [41]. In CT conditions, the clones 19 and 22 present a similar percentage of live cells than WT BV-2 (WT: 84.5 ± 6.9% of the cells, Clone 19: 83.6± 4.9%, Clone 22: 95.3 ± 0.6%) while they present lower percentages of early apoptotic (WT: 1.1 ± 0.2%, Clone 19: 1.3 ± 0.4%, Clone 22: 1.0 ± 0.1%), late apoptotic/necrotic (WT: 7.5 ± 3.2%, Clone 19: 8.0 ± 2.5%, Clone 22: 2.0 ± 0.3%) and necrotic cells (WT: 5.5 ± 2.4%, Clone 19: 7.1± 2.5%, Clone 22: 2.3 ± 0.7%) than WT BV-2. In the presence of HG conditions, both clones 19 and 22 present a similar, even slightly increased percentage of live cells than WT BV-2 (WT: 78.1 ± 9.7%, Clone 19: 81.1 ± 8.0%, Clone 22: 88.9 ± 4.2%). Similarly to CT conditions, the clones present lower percentages of early apoptotic (WT: 3.6 ± 2%, Clone 19: 1.4 ± 0.4%, Clone 22: 2.5 ± 1.1%), late apoptotic/necrotic (WT: 7.6 ± 2.8%, Clone 19: 9.4 ± 4.1%, Clone 22: 5.4 ± 2.5%) and necrotic cells than WT BV-2 (WT: 10.6 ± 5.3%, Clone 19: 7.1 ± 2.5%, Clone 22: 2.9 ± 1.1%).

To specifically evaluate proliferative activity, we therefore performed an EdU incorporation assay (Figure 5C) [42]. No significant alterations were observed either for CT or HG conditions for clones 19 and 22 in both 7 h or 12 h assays (WT 7 h HG: 99.9 ± 9.4% of WT CT; WT 12 h HG: 98.5 ± 6.2% of WT 7 h CT), (Clone 19 7 h CT: 103.2 ± 13.5% of WT 7 h CT; Clone 19 7 h HG: 96.1 ± 8.2% of WT 7 h CT; Clone 19 12 h CT: 94.9 ± 10.9% of WT 12 h CT; Clone 19 12 h HG: 90.0 ± 11.1% of WT 12 h CT), (Clone 22 7 h CT: 102.8 ± 7.7% of WT 7 h CT; Clone 22 7 h HG: 101.1 ± 9.5% of WT 7 h CT; Clone 22 12 h CT: 115.8 ± 0.4% of WT 12 h CT; Clone 22 12 h HG: 112.3 ± 4.1% of WT 12 h CT).

### 2.5. Ndr2 Downregulation Affects the Metabolism of Microglial Cells

The proliferation and apoptosis assays suggest that the decrease in the resazurin reduction by the *Ndr2* downregulated BV-2 compared to the WT BV-2 is due to metabolic changes. This led us to analyze the mitochondrial function using the Seahorse assay (Figure 6 and Appendix A).

We observed a decrease in oxygen consumption rate profile at basal level comparing the clones with the WT BV-2 cells, with a lower basal respiration (Clone 19 NG: 74.5 ± 9.8% of CT; Clone 19 HG: 69.2 ± 15.5% of CT; Clone 22 NG: 87.3 ± 14.0% of CT and Clone 22 HG: 68.5 ± 9.3% of CT), as well as maximal respiration (Clone 19 NG: 48.0 ± 20.0% of CT; Clone 19 HG: 52.7 ± 32.1% of CT; Clone 22 NG: 86.9 ± 26.2% of CT and Clone 22 HG: 80.0 ± 3.0% of CT), ATP-linked respiration (Clone 19 NG: 69.0 ± 9.0% of CT; Clone 19 HG: 43.9 ± 12.8% of CT; Clone 22 NG: 75.4 ± 29.0% of CT and Clone 22 HG: 45.0 ± 19.0% of CT) and spare respiratory capacity (Clone 19 NG: 68.0 ± 9.0% of CT; Clone 19 HG: 48.3 ± 5.0% of CT; Clone 22 NG: 64.5 ± 20.7% of CT and Clone 22 HG: 63.0 ± 2.0% of CT). We observed the expected decrease in OCR compared to WT HG with WT NG cells (basal respiration WT HG: 76.4 ± 20.8% of CT; maximal respiration WT HG: 75.8 ± 27.7% of CT; ATP-linked respiration WT HG: 65.9 ± 34.4% of CT; spare respiratory capacity WT HG: 69.7 ± 6.2% of CT) (Figure 6A, B). Moreover, we measured the glucose consumed by the WT and *Ndr2* clones BV-2 in control and HG conditions. These results demonstrate that HG negatively affects microglial cells even during acute exposures. Moreover, contrary to WT BV-2 cells, in which we observed a significant difference in glucose consumption between HG and CT conditions, the *Ndr2* downregulated BV-2 cells did not consume more glucose when exposed to a higher glycemic condition (Figure 6C). These results suggest that *Ndr2* downregulation affects the metabolism of microglial cells but does not impair their proliferation, even in HG conditions.

### 2.6. High Glucose and Ndr2 Downregulation Affect Phagocytic Efficiency

To further define how *Ndr2* downregulation affects important functions of microglial cells, we investigated the phagocytic capacity of WT and *Ndr2* downregulated BV-2 cells, in HG and CT conditions, using fluorescent yellow-green latex beads (Figure 7) [43]. WT BV-2 cells present a basal phagocytosis efficiency of 115.1 ± 18.7% (7 h assay) and 149.7 ± 8.3% (12 h assay) while the WT BV-2 cells present a phagocytosis efficiency of 105.6 ± 9.7% when exposed to HG for 7 h and of 106.2 ± 10.1% when exposed to variation in glucose. The *Ndr2* downregulated BV-2 cells exposed to CT conditions for 7 h show a non-statistically significant decrease in phagocytic efficiency compared to WT BV-2 cells (Clone 19 7 h CT: 70.6 ± 21.6%; Clone 22 7 h CT: 81.1 ± 17.8%) and HG conditions (Clone 19 7 h HG: 47.2 ± 2.3%; Clone 22 7 h HG: 87.1 ± 18.7%). For the 12 h assay, a significant decrease was observed between *Ndr2* downregulated BV-2 cells and WT cells, for both clones (Clone 19 12 h CT: 46.3 ± 18.9%; Clone 22 12 h CT: 100.8 ± 12.1%; Clone 19 12 h HG: 38.0 ± 8.7%; Clone 22 12 h HG: 95.8 ± 15.8%). Interestingly, the basal phagocytic efficiency of the WT BV-2 CT cells in the 12 h assay is higher than in the 7 h assay (WT 7 h CT: 115.05 ± 18.7%; WT 12 h CT: 149.7 ± 8.3%) suggesting the reactivity of the BV-2 cell cultures to the change in medium without alteration of the glucose levels. The basal phagocytic efficiency of WT HG is similar in both the 7 h and the 12 h assays (WT 7 h HG: 105.6 ± 9.7%; WT 12 h HG: 106.2 ± 10.1%). As an osmolarity control, we incubated cells with 25 mM of mannitol (M) along with 5.5 mM of D-glucose already present in cell culture medium during 7 h. Our results demonstrate that the cells exposed to mannitol present similar phagocytosis capability than their respective control (WT CT: 89.5 ± 14.8%; WT M: 75.7± 14.3%; 19 CT: 39.2 ± 5.3%; 19 M: 33.8 ± 1.7%; 22 CT: 23.0 ± 3.9%; 22 M: 21.7 ± 2.4%). Mannitol did not affect the phagocytosis capability of WT and *Ndr2* downregulated BV-2 cells (Appendix A) therefore the observed effects from HG exposure were not due to osmotic fluctuations.

### 2.7. High Glucose and Ndr2 Downregulation Decrease BV-2 Migration

To assess the impact of *Ndr2* downregulation on BV-2 microglia migration in CT and HG conditions, we performed the Boyden Chamber assay using transwell cell culture inserts with 8.0 µm pore diameter [44]. WT BV-2 cells and *Ndr2* downregulated BV-2 clonal populations were exposed to NG or HG for 7 h, and migrating cells, located on the bottom side of the insert, were fixed and stained with DAPI to allow cell counting. *Ndr2* downregulated BV-2 cells show a decreased migration compared to the WT parental cells, in HG conditions (Clone 19 7 h CT: 83.7 ± 17.6% of WT CT; Clone 19 7 h HG: 55.9 ± 18.3% of WT CT; Clone 22 7 h CT: 55.8 ± 19.2% of WT CT; Clone 22 7 h HG: 25.6 ± 5.2% of WT CT). Interestingly, the migration ability of the BV-2 cells exhibited a non-statistically significant decrease in cell migration in HG conditions (WT 7 h HG: 65.4 ± 9.0% of WT CT). We observed a similar effect of HG on BV-2 cells migration in *Ndr2* downregulated BV-2 cells (Appendix A).

### 2.8. Ndr2 Downregulation and Exposure to HG or LPS Upregulate the Expression of IL-17a and TNF

To determine the role of NDR2 kinase in cytokine’s secretion by microglial cells, we compared the profile of secreted cytokines by WT and *Ndr2* downregulated cells (clone 19) exposed or not to HG for 7 h, using semi-quantitative antibody microarrays. The levels of various cytokines and chemokines, such as interleukin-10 and 6 (IL-10 and-6), thrombopoietin (TPO), interleukin-17a (IL-17a), tumor necrosis factor (TNF), increased in the cell medium (CM) of clone 19 exposed to NG (19-CM CT) and exposed to HG (19-CM HG) compared with WT-CM CT and WT-CM HG, respectively. Two pro-inflammatory factors, the chemokine murine monocyte chemoattractant protein-5 (MCP5) and the soluble tumor necrosis factor receptor I (sTNFRI) were significantly decreased in the cell medium of clone 19 CT and HG compared to the cell medium of WT CT and HG, respectively (Figure 8A and Appendix A).

These results were consolidated by the assessment of the expression of TNF and IL-17 after 24 h exposure to HG or lipopolysaccharide (LPS, positive control, 100 ng/mL) by flow cytometry (Figure 8B).

For the analysis, CD11b-V500 and P2γ12-APC positive cells were selected, which correspond to microglial cells, and then the expression of IL-17a and TNF was evaluated. As we can see in Figure 8B, in the case of WT BV-2 cells, HG promoted the expression of IL-17a and TNF, when compared to the control condition. Similarly, the *Ndr2* downregulation led to an increased expression of IL-17a and TNF to similar levels as LPS, and exposure to HG did not further alter their expression levels in comparison to the control. This pattern is consistently observed in response to LPS exposure as well. In conclusion, *Ndr2* downregulation is sufficient to induce an increase in pro-inflammatory cytokines production and secretion by microglial cells, regardless of the glycose concentration in the cell culture media.

## 3. Discussion

Hyperglycemic environments are known to disrupt cellular metabolism and activate signaling pathways such as the Hippo pathway, which modulates the expression and stability of NDR kinases [13,17,45,46,47]. NDR1 and NDR2, key effectors of the Hippo pathway, are implicated in regulating cell proliferation, apoptosis, differentiation, and immune responses, including innate and antiviral defenses [14,16,18,29,48,49,50,51,52]. Our study demonstrates that NDR2 is a crucial regulator of microglial metabolic adaptation and immune functionality under hyperglycemic stress. We found that NDR2 protein levels increase in BV-2 microglial cells following high-glucose (HG) exposure and glycemic fluctuation. We also detected elevated NDR1/2 protein expression following 12 h glycemic fluctuation. These findings suggest a role for NDR kinases in the microglial response to metabolic stress. Moreover, we demonstrated that *Ndr2* downregulation impairs mitochondrial respiration, metabolic flexibility, phagocytosis, migration, and cytokine homeostasis, indicating a multifaceted role for NDR2 in maintaining microglial resilience and function in a metabolically challenging environment.

Previous studies have implicated NDR kinases in modulating inflammation [16,18,29]. We hypothesized that the increased NDR2 expression observed under HG conditions may be part of a compensatory inflammatory response to restore homeostasis. Notably, no significant change in *Ndr2* mRNA levels was observed under the same conditions, suggesting that regulation occurs at the post-transcriptional level. This may involve enhanced protein translation, increased stability, or reduced degradation under hyperglycemic stress. Further studies are warranted to clarify the mechanisms of this regulation.

To explore the functional role of NDR2 in microglial response to glucose stress, we generated *Ndr2* downregulated BV-2 clones via CRISPR-Cas9 targeting exon 7 of the *Ndr2/Stk38l* gene. Although not a complete knockout, these cells exhibited distinct phenotypes. Resazurin assays showed reduced metabolic activity in *Ndr2* downregulated clones after 7 h of HG exposure, compared to wild-type (WT) cells under control conditions. This reduction was not due to increased cell death, as confirmed by apoptosis and necrosis assays, nor due to altered cell cycle progression, as shown by EdU incorporation. These findings suggest that NDR2 is critical for sustaining cellular metabolism under stress, rather than directly regulating survival or proliferation [17,45,53].

Short-term HG exposure did not affect WT microglial viability, indicating a degree of resilience. However, prolonged HG exposure increased oxidative and nitrosative stress, leading to elevated rates of apoptosis and necrosis. These findings are consistent with literature showing that the extent and duration of hyperglycemia dictate microglial stress responses [47,54]. Seahorse analysis further revealed a significant decrease in basal oxygen consumption rate (OCR) in *Ndr2* downregulated clones, along with impaired mitochondrial respiration, including reduced maximal respiration, proton leak, and spare respiratory capacity [55,56]. Despite unaffected viability, these cells displayed lower metabolic activity, reinforcing the role of NDR2 in maintaining mitochondrial function and oxidative phosphorylation under normal and HG conditions [55,57,58]. Strikingly, while WT cells adapted to HG by reducing OCR and shifting toward glycolysis—a known metabolic reprogramming response—*Ndr2* downregulated cells failed to do so. This indicates a defect in metabolic flexibility and suggests a role for NDR2 in orchestrating energy pathway transitions. Moreover, increased proton leak and elevated complex III activity in *Ndr2* downregulated cells under HG conditions point to mitochondrial dysfunction and potential compensatory mechanisms to sustain adenosine triphosphate (ATP) production [58]. The absence of adaptive glucose consumption changes in these cells further supports impaired metabolic sensing and regulation.

Beyond energy metabolism, NDR2 appears to influence key effector functions of microglia. NDR2 colocalized with ionized calcium-binding adapter molecule 1 (IBA1), an actin-crosslinking protein critical for membrane ruffling and phagocytosis [32,59]. Localization of NDR2 to membrane ruffles and process tips suggests a role in cytoskeletal remodeling. Consistently, *Ndr2* downregulated cells exhibited reduced phagocytic activity under both control (CT) and HG conditions, likely due to a combination of energy deficits and actin cytoskeletal dysfunction. Microglial migration, assessed using a Boyden chamber assay, was similarly impaired in *Ndr2* downregulated cells, again under both CT and HG conditions. Migration relies on ATP-dependent actin remodeling and purinergic signaling [60]. Functionally, these impairments in phagocytosis and migration were time- and glucose-dependent. Phagocytosis was significantly reduced under 7 h and 12 h HG exposure in *Ndr2* downregulated cells, whereas WT cells maintained or even enhanced phagocytic capacity under similar conditions. This underscores the role of NDR2 in facilitating adaptive responses to metabolic stress and maintaining microglial functionality.

Importantly, although *Ndr2* downregulated cells remained viable and proliferative, their impaired functional outputs suggest that NDR2 selectively regulates energy-dependent immune functions rather than general cell survival.

Microglia are metabolically flexible cells, that normally rely on oxidative phosphorylation (OXPHOS) under homeostatic conditions but shift toward glycolysis when activated [31,61,62]. Recent studies have highlighted the intricate relationship between microglial function and metabolism, showing that metabolic reprogramming is a key driver of inflammatory phenotypes. This transition resembles aerobic glycolysis (the “Warburg effect”), classically described in activated immune cells and tumor biology, and is increasingly recognized as a hallmark of microglial [31,61,63,64,65]. Glycolysis, culminating in the pyruvate kinase–mediated conversion of phosphoenolpyruvate to pyruvate and ATP, provides rapid energy but also sustains inflammatory signaling by facilitating cytokine and reactive oxygen species generation through ancillary pathways [64]. Indeed, during neuroinflammation or injury, microglia adopt glycolysis to meet increased energy demands, and inhibiting glycolysis dampens their activation, further linking metabolism to immune function [31,66]. Maintaining equilibrium between OXPHOS and glycolysis is therefore critical for balancing inflammatory and homeostatic microglial functions. For example, Baik et al. showed that in vitro exposure to Aβ plaques induces a metabolic shift from OXPHOS to aerobic glycolysis, resulting in increased pro-inflammatory cytokine release and enhanced phagocytosis via the mTOR–HIF-1α pathway [67].

Beyond microglia, metabolic reprogramming is also evident in astrocytes and neurons, where enhanced glycolytic activity supports physiological neuronal function but can exacerbate inflammatory cascades when excessive [68]. Collectively, such metabolic shifts have been implicated in the pathogenesis of a wide range of neurodegenerative diseases, including Alzheimer’s disease, multiple sclerosis, Parkinson’s disease, amyotrophic lateral sclerosis, Huntington’s disease, and diabetic retinopathy.

Together, these findings reinforce the concept that the balance between OXPHOS and glycolysis dictates microglial phenotype and function across neurological disorders. Our results extend this paradigm by positioning NDR2 as a novel upstream regulator of microglial metabolic adaptability under hyperglycemia, linking impaired metabolic flexibility to mitochondrial dysfunction, dysregulated cytokine production, and ultimately neurovascular pathology in diabetic retinopathy. Consistent with this, we found that Ndr2 downregulation induced a dysregulated inflammatory phenotype, particularly under high-glucose (HG) conditions. Ndr2-deficient clones secreted higher levels of pro-inflammatory cytokines, including tumor necrosis factor (TNF), interleukin-6 and 17a (IL-6 and IL-17), even in the absence of lipopolysaccharide (LPS) stimulation. Interestingly, some cytokines such as RANTES (regulated upon activation, normal T cell expressed and presumably secreted), murine monocyte chemoattractant protein-5 (MCP-5), and soluble tumor necrosis factor receptor I (sTNFRI) were reduced, suggesting an imbalanced or mixed activation profile. The decrease in sTNFRI may reduce the buffering of TNF signaling, thereby enhancing its cytotoxic effects. Meanwhile, compensatory upregulation of IL-10 and IL-13 likely reflects an attempted resolution of inflammation [69,70,71,72,73]. Overall, the cytokine profile observed—characterized by elevated IL-6, IL-2, IL-12, IL-17, IL-9, GM-CSF (granulocyte-macrophage colony-stimulating factor), TNF, interferons (IFNs), and MCP-1, alongside reduced granulocyte colony-stimulating factor (G-CSF), vascular endothelial growth factor (VEGF), IL-4, sTNFRI, and MCP-5—indicates an aberrant activation state that promotes immunostimulation and endothelial activation [63,69,73,74,75,76,77]. In turn, this may drive upregulation of adhesion molecules, leukocyte recruitment, and vascular permeability, thereby facilitating immune cell infiltration and exacerbating tissue injury.

Accordingly, *Ndr2* downregulation and/or deletion may promote inflammation and blood–brain/retina barrier dysfunction, contributing to neurodegenerative diseases such as diabetic retinopathy, Parkinson’s or Alzheimer’s disease. In addition, cytokines including IL-10, IL-6, and TNF regulate synaptic pruning, elimination, and plasticity; their imbalance may therefore cause excessive synapse loss or impaired circuit refinement, ultimately leading to cognitive decline or visual dysfunction [78].

This complex cytokine signature is characteristic of chronic inflammatory microglial states and may reflect aberrant activation of nuclear factor kappa-light-chain-enhancer of activated B cells (NF-κB) and mitogen-activated protein kinase (MAPK) pathways, which are master regulators of stress responses, immune activation, and cell survival, and are frequently upregulated in diabetic retinopathy, age-related macular degeneration, and other retinal pathologies [79,80,81]. Previous studies have shown that NDR2, as well as its paralog NDR1, directly and indirectly modulates NF-κB and MAPK signaling, thereby shifting microglia toward a pro-inflammatory state and promoting vascular dysfunction at both microglial and endothelial levels. While targeting NDR2 may help restore NF-κB/MAPK balance in retinal disease, such interventions must be carefully evaluated given the risk of systemic immune dysregulation.

Collectively, these findings indicate that NDR2 is a central regulator of microglial inflammatory homeostasis. In the context of diabetic retinopathy, *Ndr2* downregulation may represent a key mechanism linking hyperglycemia to early neurovascular dysfunction. Microglial activation is an established driver of retinal inflammation in DR, and our results suggest that loss of NDR2 amplifies this process by fostering a pro-inflammatory cytokine milieu, impairing mitochondrial function, and reducing metabolic adaptability. This imbalance exacerbates oxidative stress and sustains NF-κB/MAPK activation, thereby fueling endothelial activation, leukocyte recruitment, and breakdown of the blood–retina barrier. Importantly, these alterations align with early pathogenic events in DR, when microglial dysfunction and vascular instability preceded over neovascularization. Thus, NDR2 downregulation may act as an upstream regulator of microglial pathogenicity under hyperglycemia, offering a potential therapeutic entry point to prevent or slow the progression of DR.

Beyond DR, *Ndr2* downregulation mirrors features of chronic neuroinflammation characteristic of retinal diseases such as age-related macular degeneration, as well as brain disorders including Parkinson’s disease, Alzheimer’s disease, and autoimmune neurodegeneration.

A major strength of this study is the use of physiologically relevant glucose exposure time points—specifically 7 h and 12 h conditions—to model both acute hyperglycemia and stress induced by daily glycemic variability. This approach more accurately reflects the transient but impactful glucose fluctuations observed in individuals with diabetes and metabolic disorders, thereby enhancing the translational value of our findings.

Nonetheless, our study has several limitations. The use of *Ndr2* downregulation, rather than a complete knockout, may limit the interpretation of the full functional spectrum of NDR2. Additionally, although BV-2 cells are a widely accepted model for microglia, they may not fully recapitulate the behavior of primary microglia within the central nervous system microenvironment. Primary or iPSC-derived microglia would provide a model more representative of *in vivo* physiology. However, isolating primary retinal microglia [82] or generating iPSC-derived microglia yields a limited number of cells [83], which constrain their systematic use in large-scale functional assays such as those performed here. Given these practical considerations, we employed the BV-2 cell line, a well-established and validated in vitro model for studying inflammatory mechanisms and microglial responses, which allowed us to ensure reproducibility and consistency. Importantly, to strengthen the robustness of our study, we also utilized primary and iPSC-derived microglial cultures to characterize NDR2 localization, thereby reinforcing the biological relevance of the findings obtained in BV-2 cells. Finally, it remains to be determined which direct downstream targets of NDR2 mediate its regulatory effects on mitochondrial function and cytoskeletal dynamics.

To strengthen translational relevance, in vivo studies are needed to validate these findings in a more physiologically and pathologically accurate context. These studies should also examine whether modulating NDR2 can reverse established inflammatory phenotypes or neurodegenerative changes. Furthermore, future investigations should explore whether other components of the Hippo signaling pathway similarly influence microglial metabolism and function, potentially unveiling broader therapeutic targets for neuroinflammatory and metabolic diseases.

## 4. Materials and Methods

### 4.1. Animal Care

All procedures involving animals were approved by the Animal Welfare Committee of the Faculty of Medicine of University of Coimbra (ORBEA 9-2022, 12/01/2022) and conducted in accordance with the European Community directive guidelines for the use of animals in laboratory (2010/63/EU) transposed to the Portuguese law in 2013 (Decreto-Lei 113/2013), and in agreement with the Association for Research in Vision and Ophthalmology statement for animal use.

### 4.2. Mouse Retinal Microglial Cell Culture

Primary retinal microglial cell cultures were obtained from 3 to 4-day-old C57BL/6J mice pups, as described previously [43,84], with some modifications, as follows. The retinas were dissociated by trypsinization (0.25% trypsin supplemented with 0.05 mg/mL DNase I) followed by mechanical dissociation. The cell suspension was plated in uncoated T75 flasks (corresponding to eight retinas per flask) and maintained in Dulbecco’s Modified Eagle Medium/Nutrient Mixture F-12 (DMEM/F-12) with GlutaMAX™ (Gibco, Thermo Fisher Scientific, Waltham, MA, USA) supplemented with 10% Fetal Bovine Serum (FBS, Gibco, Thermo Fisher Scientific, Waltham, MA, USA), 100 U/mL penicillin, 100 μg/mL streptomycin (Gibco, Thermo Fisher Scientific, Waltham, MA, USA) and 50 ng/mL of recombinant mouse CSF1 (#78059.1, Stemcell, Vancouver, BC, Canada). A mild trypsinization (0.25% trypsin, Thermo Fisher Scientific, Waltham, MA, USA) was performed every 3-4 days to remove active microglia cells and prevent cytokine overload of the culture. When the culture reached near confluency (~2 weeks), cells were trypsinized and plated at a density of 0.5  ×  104 cells/well in µ-Slide 8 Well high (#80801, Ibidi, Fitchburg, WI, USA) coated with poly-D-lysine (0.1 mg/mL, #A3890401, Gibco, Thermo Fisher Scientific, Waltham, MA, USA) and cultured at 37 °C in a humidified atmosphere of 5% CO_2_. The purity of the culture (>90%) was assessed by immunocytochemistry with anti-Iba1 antibody.

### 4.3. Human iPSC-Derived Microglial Cell Culture

Macrophage/microglia progenitors were provided by Dr. Ana Luisa Cardoso (CNC, UC) and prepared as described by Haenseler et al. [83]. Briefly, approximately 4 weeks after plating embryoid bodies in T175 flasks in the presence of X-VIVO15 medium (Lonza, Basel, Switzerland), supplemented with 100 ng/mL M-CSF (Invitrogen, Waltham, MA, USA), 25 ng/mL IL-3 (R&D Systems, Minneapolis, MN, USA), 2 mM Glutamax (Invitrogen, Waltham, MA, USA), 100 U/mL penicillin and 100 mg/mL streptomycin (Invitrogen, Waltham, MA, USA), and 0.055 mM b-mercaptoethanol (Invitrogen, Waltham, MA, USA), with fresh medium added weekly, macrophage/microglia progenitors start to emerge in the supernatant. These cells were collected by medium centrifugation and further differentiated into iPSC-derived microglial cultures. For that purpose, cells were counted and seeded at 3.0 × 10^4^ cells/well in fibronectin-coated Ibidi µ-Slide 8 Well plates, in DMEM/F12 medium supplemented with N2, 100 ng/mL IL-34 (R&D Systems, Minneapolis, MN, USA) and 10 ng/mL GM-CSF (R&D Systems, Minneapolis, MN, USA). The microglial progenitors were matured for 2 weeks, with medium changes 3 times per week, before being used for experiments.

### 4.4. BV-2 Cell Culture

BV-2 microglial cells, obtained from the laboratory of Dr. António Francisco Ambrósio (University of Coimbra) were cultured in Roswell Park Memorial Institute 1640 culture medium (RPMI; #11875093, Gibco, Thermo Fisher Scientific, Waltham, MA, USA) supplemented with 10% (*v*/*v*) FBS, 1% (*v*/*v*) L-glutamine (#25030024, Thermo Fisher Scientific, Waltham, MA, USA), and 1% (*v*/*v*) Penicillin-Streptomycin (Pen/Strep; #15140122, Thermo Fisher Scientific, Waltham, MA, USA). Cells were then maintained in RPMI at 37 °C in a humidified atmosphere with 5% CO_2_ and kept until passage 27 (P27). All cell lines were tested for mycoplasma contamination using PCR-based assays and confirmed negative prior to use in experiments. Each *n* corresponds to data obtained from an independent culture in a separate experiment, except where otherwise stated.

### 4.5. High-Glucose Assay

BV-2 cells were seeded at a density of 1.0 × 10^4^ cells/cm^2^ in Dulbecco’s Modified Eagle Medium low glucose (DMEM NG; #31600083, Gibco, Thermos Fisher Scientific, Waltham, MA, USA) containing 5.5 mM glucose (1 g/L) and supplemented with 10% (*v*/*v*) FBS, 2 mM L-glutamine, and 1% (*v*/*v*) Pen/Strep. This condition is called normal glucose condition (NG) or control condition (CT). After 16 h–18 h, in some of the wells, the culture medium was supplemented with 25 mM D-glucose (#G8270, Sigma-Aldrich, St. Louis, MO, USA) in addition to the 5.5 mM already present in DMEM, reaching a final concentration of 30.5 mM (HG; 5 g/L) to simulate hyperglycemic conditions observed in diabetes.

The cells were exposed to HG for 7 h (7 h assay) or two times 4 h with an interval of 4 h between exposures in HG (12 h assay). Then, BV-2 were kept in normal DMEM until the next morning for subsequent experiments (Figure 2).

### 4.6. Plasmids, BV-2 Transfection and Clonal Selection

Plasmid. *Ndr2/Stk38l* clonal downregulated BV-2 cell lines were generated by introduction of frameshift mutations in *Ndr2* exon 7 by CRISPR/Cas9 gene-targeting methods, as described [85]. The empty plasmid pX459, catalog number #62988, containing a puromycin resistance cassette and Cas9 nuclease, was obtained from Addgene. The all-in-one plasmid used for genome editing was obtained from Vector Builder (Neu-Isenburg, Germany) and contains one sgRNA targeting the *Ndr2* exon 7 (Table 2), designed using the CRISPR design tool CHOPCHOP version 3.0.0 (https://chopchop.cbu.uib.no/ (accessed on 11 March 2023)) [33,34] and inserted in the pRP[CRISPR]-Puro-hCas9-U6 vector allowing the co-expression of a puromycin resistance cassette and Cas9 nuclease. The vector ID is VB230329-1589djw, which can be used to retrieve detailed information about the vector on vectorbuilder.com (https://en.vectorbuilder.com/ (accessed on 28 March 2023)).

The specificity of CRISPR/Cas9 editing depends on the sgRNA sequence and on the presence of a protospacer adjacent motif located next to the target sequence. The sgRNA sequence was chosen to target *Ndr2* exon 7 coding for the kinase domain while exhibiting the highest quality score by inverse likelihood of off-target binding based on the CRISPR design tool developed by Zhang Lab, MIT 2017 (https://zlab.squarespace.com/guide-design-resources (accessed on 22 October 2025)) [85].

For off-target prediction, off-target site predictors CRISPOR version 5.2 was employed (Appendix A (https://crispor.gi.ucsc.edu/ (accessed on 3 September 2025)) [35,36,37].

Transfection. BV-2 cells were seeded into a T25 flask, at a density of 2.0 × 10^4^ cells/cm^2^ with 5 mL of RPMI medium and then maintained at 37 °C in a humidified atmosphere with 5% CO_2_ for 18 h. The following day, transfection with Lipofectamine 3000 (#L3000001, Thermo Fisher Scientific, Waltham, MA, USA) was performed on cells (70–90% confluency) for 8 h in Opti-MEM (#31985-047, Gibco, Thermo Fisher Scientific, Waltham, MA, USA), after which Opti-MEM was changed to RPMI.

Selection and isolation of clonal cell lines. The morning of the next day, puromycin (3 µg/mL) was added to the transfected cells and kept for 30 h to select the cells that had successfully incorporated the plasmid of interest. Once the cells reached a confluence of approximately 70%, the transfected cells were diluted to 100 cells in 12 mL of RPMI and seeded into a 96-well plate to obtain multiple clones that could be further expanded. 10 colonies (round and radiating from a central point) were selected and allowed to expand for 2–3 weeks. The puromycin concentration for the selection of the transfected BV-2 cells was established by performing resazurin assay after 24 h, 30 h and 48 h exposure to 2, 3 and 5 µg/mL of puromycin. For this test, we used the pX459 plasmid given to us by Christopher J. Lengner (School of Veterinary Medicine at the University of Pennsylvania) (pSpCas9(BB)-2A-Puro (PX459) V2.0 was a gift from Feng Zhang, Addgene plasmid # 62988; http://n2t.net/addgene:62988 (accessed on 3 September 2025); RRID:Addgene 62988), carrying a puromycin-resistance cassette [85].

### 4.7. Ndr2/Stk38l CRISPR BV-2 Cells Validation

DNA was isolated from BV-2 cells using ethanol precipitation. Sanger sequencing was used to identify insertion/deletion (Indels) caused by non-homologous end joining repair at the cut site in *Ndr2* exon 7 (see Table 2 for oligos). To validate *Ndr2* deletion and compare relative *Ndr2* transcript levels in WT and the selected clones (clone 8, 13, 19, and 22), quantitative RT-PCRs (RT-qPCR) were conducted to probe mouse cDNA for transcripts that span *Ndr2* exons 13–14 (see Table 2 for oligos).

### 4.8. Quantitative RT-qPCR

Total RNA was extracted using a column-based RNA purification kit (Zymo Quick-RNA MiniPrep Kit, #R1055, Orange, CA, USA), following the manufacturer’s protocol for adherent cells. For cDNA synthesis, samples were considered suitable when the ratio values for both A260/A280 and A260/A230 were within the range of 1.8 to 2.2. cDNA was synthesized using the NZY First-Strand cDNA Synthesis Kit (Nzytech, Lisboa, Portugal), following the manufacturer’s instructions. The resulting cDNA was diluted to a final concentration of 25 ng/µL. RT-qPCR experiments were done in compliance with standard MIQE (Minimum Information for Publication of Quantitative Real-Time PCR Experiments) guidelines [86]. The RT-qPCR reactions contained 50 ng cDNA, 1× iTaq Universal SYBR Green mix (#r1725122, Bio-Rad, Hercules, CA, USA) and 250 nM of each unlabeled forward and reverse primer. Reactions were performed in 96-well reaction plates using the StepOnePlus PCR System (Applied Biosystems, Waltham, MA, USA). Ywhaz was found to be the most stable housekeeping gene and used for normalization and calculation using the ΔΔCT method [86]. Primers for RT-qPCR are listed in Table 2.

### 4.9. SDS-PAGE and Western Blot

Cells from at least 5 independent cultures were washed three times with ice-cold phosphate-buffered saline (PBS) and then homogenized in ice-cold RIPA lysis buffer (50 mM Tris HCl, pH 7.4, 150 mM NaCl, 5 mM ethylenediamine tetra-acetic acid (EDTA), 1% Triton X-100, 0.5% sodium deoxycholate, 0.1% sodium dodecyl sulfate (SDS), 1 mM dithiothreitol supplemented with 10% protease inhibitors (Roche Complete Mini-EDTA free protease inhibitors, # COEDTAF-RO, Roche, Basel, Switzerland).

Proteins (20 µg of total protein loaded) were separated on a 12% SDS-polyacrylamide gel and transferred onto a PVDF membrane. Membranes were blocked in 5% (*w*/*v*) low-fat dry milk/Tris-buffered saline with 0.1% (*v*/*v*) Tween 20 (TBS-T) for 1 h at room temperature (RT), treated with primary antibody in blocking buffer overnight (O/N) at 4 °C and incubated for 1 h at RT in the secondary antibody. Chemiluminescence was detected using the western-bright ECL kit (#K-12043-D20, Advansta, San Jose, CA, USA). Band intensity was quantified through Image Lab Software version 6.1.0 (Bio-Rad, Hercules, CA, USA) and normalized to the intensity of loading protein (calnexin). All commercially available antibodies used for immunoblots and immunofluorescence experiments are listed in Table 3.

### 4.10. Alamar Blue Assay

Twenty-four hours after starting the HG assays, cell viability was assessed using resazurin dye (7 hydroxy-3H-phenoxazin-3-one 10-oxide; stock at 500 µM), diluted in PBS at a final concentration of 50 µM. Following a 4 h incubation at 37 °C and 5% CO_2_, the absorbance at 570 nm and 620 nm was measured by a multimode microplate reader (Biotek Synergy HT, Winooski, VT, USA) to assess the conversion of resazurin to fluorescent resorufin. By analyzing the relative absorbance values, proportional to the amount of metabolically active live cells present in the culture, the percentage of cell viability and the metabolic status of BV-2 cultures can be determined, as follows:% viability = (Absorbance of sample)/(Absorbance of control) × 100

### 4.11. Immunocytochemistry

Cells were washed with PBS and fixed with 100% methanol at −20 °C for 10 min at RT. Subsequently, the cells were permeabilized and blocked with 3% bovine serum albumin (BSA) and 0.25% Triton X-100 in PBS for 30 min, treated with the primary antibody in blocking buffer O/N at 4 °C and incubated for 1 h at RT in secondary antibody and 0.5 µg/mL DAPI (#D1306, Invitrogen, Waltham, MA, USA).

### 4.12. Microscopy and Imaging/Quantification

Immunofluorescence microscopy as conducted using either a widefield imaging microscope (Zeiss Axio Observer.Z1 inverted microscope equipped with a Colibri 7 LED light source, CCD digital camera (AxioCam HRm) and Zen Blue 2012 software) with a 20× (Plan-Apochromat, 0.8 NA) or an 40× (LD Plan-Neofluar, 0.6 NA Corr) objective, or a laser scanning confocal microscope (LSM 710 Axio Observer.Z1 microscope with QUASAR detection unit and Zen Black 2010 software) with a 63× (Plan-Apochromat, 1.4 NA oil) objective. All the equipment was from Carl Zeiss (Carl Zeiss, Oberkochen, Germany). All experimental conditions within a particular experiment were processed simultaneously, and imaging settings (exposure time and laser power) were conserved. An average of 5 images per coverslip were selected to quantify NDR2, and analyzed using ImageJ v.1.54p normalized to the number of cells per image, and the mean fluorescence intensity from these five images was used to determine the result for each condition.CTCF = Integrated Density − (Area of selected cells × Mean fluorescence background)

### 4.13. Phagocytic Activity Assay

(#L1030-0001, Sigma-Aldrich, St. Louis, MO, USA) were added to the cell medium to reach a final concentration of 0.0025% and incubated for 1 h at 37 °C with 5% CO_2_. The cells were washed three times with PBS to remove the non-phagocytized beads and fixed with 100% methanol at −20 °C for 10 min. BV-2 cells were stained with TRITC-conjugated phalloidin (1:500; #P1951, Sigma-Aldrich, St. Louis, MO, USA) and nuclei were stained with 0.5 µg/mL DAPI (#D1306, Invitrogen, Waltham, MA, USA). Cells were observed with a Carl Zeiss widefield microscope (Zeiss Axio Observer.Z1 inverted microscope, Carl Zeiss, Oberkochen, Germany), and five random fields were acquired from each condition. The analysis was performed through ImageJ. The phagocytic efficiency was calculated based on the following formula, with x being the number of beads [87]% Phagocytic Activity = (1*1x + 2*2x + ⋯ + n*nx)/(Total Number of Cells) × 100

### 4.14. Boyden Chamber Assay

24 h prior to the assays in which BV-2 cells were exposed to glucose, the medium of BV-2 cells was replaced with DMEM containing 0% FBS. The next day, BV-2 cells were seeded into Transwell cell culture inserts (8 µm pore diameter, Millipore, Burlington, MA, USA) at a density of 1 × 10^4^ cells/cm^2^ in either NG or HG conditions. At the end of the experiments, cells were fixed in 4% PFA with 4% sucrose, and after 15 min, cells on the upper side of the transwell were removed with a cotton swab. Nuclei were stained with DAPI (1:2000) for 20 min to allow cell counting. The membranes were carefully removed from the inserts with a blade and mounted on glass slides with Dako Fluorescent Mounting Medium (#S302380-2, Agilent, Santa Clara, CA, USA). The samples were observed in an inverted fluorescence microscope (Zeiss Axio Observer.Z1 inverted microscope, Carl Zeiss, Oberkochen, Germany) and five fields were acquired from each membrane. The number of migrated cells was counted.

### 4.15. Proliferation Assay (EdU)

Cell proliferation was evaluated using the EdU staining proliferation kit (#ab222421, Abcam, Cambridge, UK), following the manufacturer’s protocol. EdU was added to the cells at a concentration of 10 µM Five random fields were acquired from each coverslip, the number of Edu-positive cells was counted, and the results were normalized to the total number of microglia. The results are presented as the percentage of the WT control.% Proliferation = (Number of EdU Positive Cells)/(Total Number of Cells) × 100

### 4.16. Flow Cytometry

Cell death assay. Cell death was assessed using flow cytometry through the double staining with annexin V (AV) and 7-aminoactinomycin D (7-AAD, BioLegend, San Diego, CA, USA). After HG exposure, the cells were washed with PBS by centrifugation at 400× *g* for 5 min. After that, the cells were stained with AV and 7-AAD and analyzed as described [88]. Briefly, 1.0 × 106 cells were collected and washed with PBS, centrifuged at 500× *g* for 5 min, resuspended in 100 μL of AV binding buffer and incubated with 2.5 μL of AV and 5 μL of 7-AAD for 15 min in the dark at RT. Then, cells were diluted in 300 µL of AV binding buffer and analyzed in FACSCalibur flow cytometer (BD Pharmingen, Franklin Lakes, NJ, USA). At least 25,000 events were acquired using CellQuest software v.1.0.2 (BD Pharmingen, Franklin Lakes, NJ, USA) and analyzed using Kaluza Analysis Software v. 2.4 (Beckman Coulter, Brea, CA, USA). Results represent the mean ± SEM of percentage of each cell population for 3-4 independents experiments: viable cells (AV−/7-AAD−), initial apoptotic (AV+/7-AAD−), late apoptotic/necrotic (AV+/7-AAD+), and necrotic cells (AV−/7-AAD+).

Cytokines. Cells (5 × 10^5^) were seeded into a 10 cm dish and exposed to HG or lipopolysaccharide (LPS, 100 ng/mL) for 24 h. After 20 h, 5 µL of Brefeldin A (Sigma-Aldrich, St. Louis, MO, USA) was added to the culture to inhibit the secretion of cytokines and allow better detection by flow cytometry. Following this, cells were gently detached, centrifuged for 5 min at 230× *g*, and resuspended in PBS for flow cytometry analysis.

For surface marker detection, cell pellets were incubated with fluorescent conjugated antibodies against CD11b-V500 (clone M1/70, RRID:AB_398535, BD Pharmingen, Franklin Lakes, NJ, USA), P2γ12-APC (clone S16607D, RRID:AB_2721468, BioLegend, San Diego, CA, USA), TNFa-PE (clone TN3-19.12, RRID:AB_315418, BioLegend, San Diego, CA, USA), and IL-17a-V450 (clone N49-653, RRID:AB_1727539, BD Pharmingen, Franklin Lakes, NJ, USA). The control conditions include unstained cells, and single labeling controls compensation of the flow cytometer. The work conditions were categorized into two tube groups: (i) extracellular labeling without permeabilization (CD11b-V500, P2γ12-APC) and (ii) combined extracellular and intracellular labeling with permeabilization. The antibodies for extracellular labeling were added in all tubes, thoroughly mixed, and then incubated for 15 min in the dark at RT. After incubation, one PBS wash was performed, and cells of the first group were resuspended in PBS for flow cytometry reading. For tubes of the second category, 100 µL of Fix and Perm A solution (Invitrogen, Waltham, MA, USA) was added. After 10 min incubation, one PBS wash was performed followed by 20 min incubation in 100 µL of the intracellular antibodies (TNFa-PE, IL-17a-V450) diluted in Fix and Perm B solution (Invitrogen, Waltham, MA, USA), in the dark and at RT. After incubation, cells were washed twice with PBS buffer and acquired in an 8-color flow cytometer, BD FACSCanto II (BD Pharmingen, Franklin Lakes, NJ, USA). Data analysis was performed using FlowJo v.10.7 (BD Pharmingen, Franklin Lakes, NJ, USA).

### 4.17. Seahorse Analysis

Oxygen consumption rate (OCR) was measured at 37 °C using an XF24 extracellular flux analyzer (Seahorse Bioscience, Billerica, MA, USA). Briefly, the cells were seeded in an Agilent Seahorse 24-well XF culture microplate at a density of 0.6 × 104 cells/well in 250 µL of DMEM NG, allowed to adhere for 24 h in a humidified incubator with 5% CO_2_ at 37 °C. Cells were subsequently treated with 500 µL of DMEM HG or DMEM NG for 7 h, and then cultured in 500 µL of DMEM NG until the following day. The day prior to the Seahorse experiment, the Seahorse XFSensor Cartridge was hydrated and calibrated with 1 mL of Seahorse XF Calibrant Solution in a non-CO2 37 °C humidified incubator.

Mitochondrial functionality of BV-2 cells was analyzed using 1.5 µM oligomycin (#O4876, Sigma-Aldrich, St. Louis, MO, USA), 5 µM of BAM15 (Sigma-Aldrich #SML1760), 0.5 µM of rotenone (#R8875, Sigma-Aldrich, St. Louis, MO, USA) and 1 µM of antimycin A (#A8674, Sigma-Aldrich, St. Louis, MO, USA). Each experiment was performed for at least 3 independent biological replicates. Data was normalized based on protein content using the sulforhodamine B (SRB) assay [89].

### 4.18. Mouse Cytokine Antibody Array

Mouse cytokine antibody arrays (Membrane, 22 Targets, #ab133993, Abcam, Cambridge, UK) were used to profile the following cytokines secreted in the medium of WT and *Ndr2* downregulated BV-2 cells, in NG and HG conditions: G-CSF, GM-CSF, IL-2, IL-3, IL-4, IL-5, IL-6, IL-9, IL-10, IL-12 p40/p70, IL-12p70, IL-13, IL-17, IFN-gamma, MCP-1, MCP-5, RANTES, SCF, sTNFRI, TNF, Thrombopoietin or TPO, VEGF. Each sample is the pool of *n* = 3–6 supernatants, vol/vol. The experiment was carried out according to the manufacturer’s instructions. Briefly, antibodies-coated membranes were first incubated for 30 min with 2 mL of blocking buffer. After 30 min, the blocking buffer was replaced with 1 mL supernatant from WT 7 h CT, WT 7 h HG, Clone 19 7 h CT and Clone 19 7 h HG BV-2 cells samples. Membranes were incubated overnight at 4 °C with mild shaking. The next day, the membranes went through a large volume wash, 3 washes with the Wash Buffer I and 2 washes with the Wash Buffer II before being incubated with 1 mL biotin-conjugated antibodies overnight at 4 °C with mild shaking. Lastly, the membranes were washed as before and incubated with HRP-conjugated streptavidin (overnight at 4 °C with mild shaking), then revealed (<10 min) using a chemiluminescence substrate monitored on an Image Lab Software v.6.1.0 (Bio-Rad, Hercules, CA, USA).

## 5. Conclusions

Our findings position NDR2 as a key regulator of microglial mitochondrial function, metabolic flexibility, and effector responses such as phagocytosis and migration. While short-term glycolytic activation supports microglial responses, chronic reliance on glycolysis—especially in the context of impaired mitochondrial function—leads to pro-inflammatory bias, reduced functional plasticity, and diminished debris clearance. *Ndr2* downregulation recapitulates this phenotype, highlighting its role in metabolic adaptation and immune regulation.

Given the prevalence of metabolic dysregulation in neurodegenerative and metabolic diseases, NDR2 represents a promising therapeutic target for restoring microglia homeostasis and mitigating chronic inflammation. Future research should explore the molecular pathways downstream of NDR2 that govern metabolic reprogramming, cytoskeletal dynamics, and cytokine signaling in microglia.

## Figures and Tables

**Figure 1 ijms-26-10630-f001:**
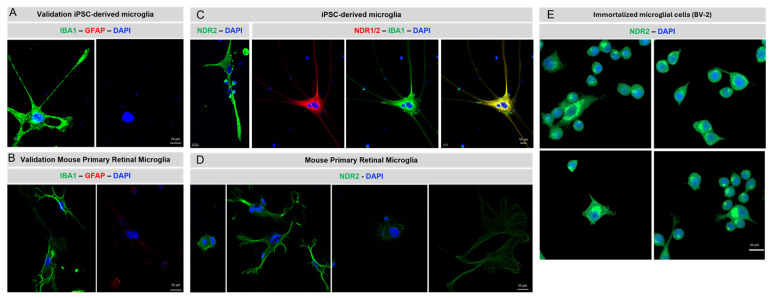
NDR2 expression in human and mouse microglial cells. (**A**,**B**) The microglia phenotype was validated in both human iPSC-derived microglial cultures and mouse primary microglial cultures by immunocytochemistry. IBA1: microglial cell marker (green); GFAP: astrocyte and reactive Müller cell marker (red). (**C**–**E**) Immunocytochemistry against NDR2 (green) in human iPSC-derived microglial cultures, mouse primary microglial cultures, and BV-2 cell cultures. DAPI: nuclei staining (blue). Scale bar: 10 µm for both human iPSC-derived microglial cultures and mouse primary microglial cultures, 20 µm for BV-2.

**Figure 2 ijms-26-10630-f002:**
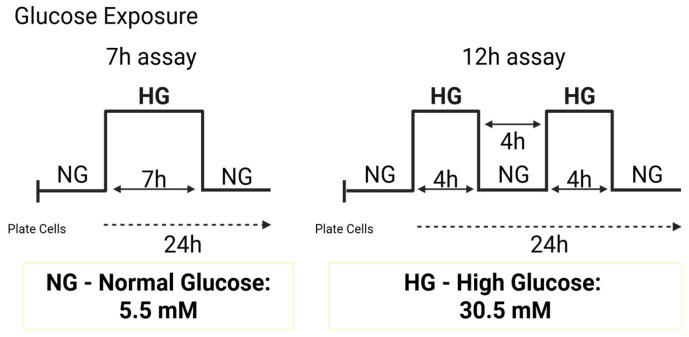
Schematic representation of the different experimental protocols used for exposure to high-glucose (HG) levels. BV-2 cell cultures were incubated in 5.5 mM glucose (normal glucose; NG or CT) or 30.5 mM glucose (high-glucose; HG) for different periods: 7 h (7 h assay) or two times 4 h HG with a break of 4 h in between in NG (12 h assay) (Created with BioRender.com).

**Figure 3 ijms-26-10630-f003:**
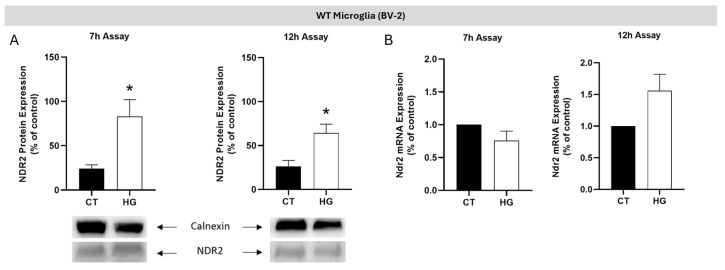
High-glucose upregulates the NDR2 protein expression in BV-2 microglial cells. (**A**) NDR2 protein levels were assessed by Western blot in WT BV-2 cells exposed to CT or HG for the 7 h and 12 h assays. Results are expressed as the ratio of NDR2 to Calnexin intensity, normalized to CT (arbitrary units, a.u.) ± SEM. Statistical analysis was performed using Student’s *t*-test after confirming Gaussian distribution; *p* ≤ 0.05: HG vs. CT; *n* = 6–7. (**B**) *Ndr2* mRNA levels in BV-2 cells under the same glucose conditions and time points were evaluated by qRT-PCR. Results are expressed as percentage of CT ± SEM. Statistical analysis was performed using Student’s *t*-test * *p* ≤ 0.05, compared to WT CT; *n* = 5.

**Figure 4 ijms-26-10630-f004:**
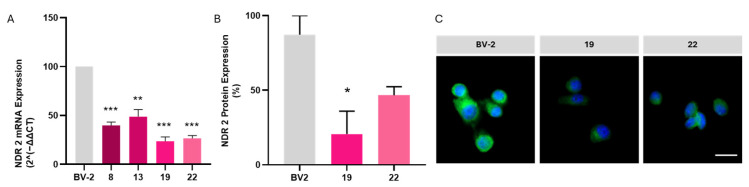
*Ndr2* downregulated BV-2 clones. (**A**) *Ndr2* mRNA levels were evaluated by qRT-PCR in WT (BV-2) and in four *Ndr2* downregulated BV-2 clones (Clones 8, 13, 19, and 22). Results are expressed as a percentage of WT ± SEM (*n* = 2). Statistical analysis was performed using one-way ANOVA; *p* ≤ 0.05, *p* ≤ 0.005, *p* ≤ 0.001, compared with WT. (**B**,**C**) NDR2 protein expression was assessed by immunocytochemistry (green, in (**C**)) in WT cells and *Ndr2* downregulated clones (Clone 19 and Clone 22). DAPI: nuclei staining (blue). Results in B are expressed as a percentage of WT based on mean fluorescence intensity (MFI). Statistical analysis was performed using one-way ANOVA; * *p* ≤ 0.05, ** *p* ≤ 0.01, *** *p* ≤ 0.001compared with BV-2 WT (named BV-2). Data represents two independent cultures (*n* = 2), with 40 cells individually analyzed per culture. Scale bar: 20 µm.

**Figure 5 ijms-26-10630-f005:**
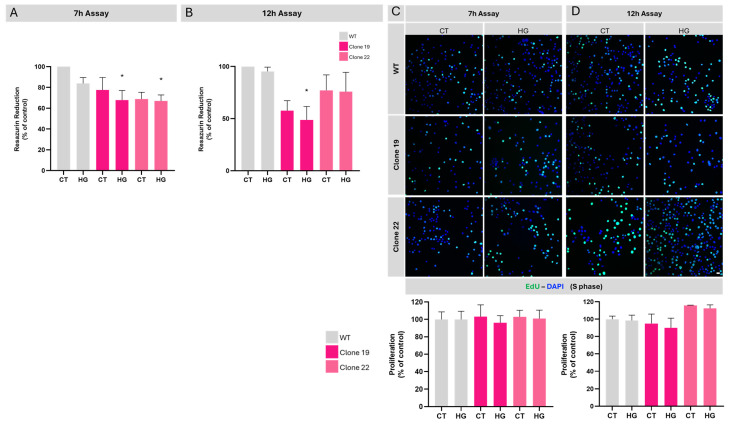
Viability and proliferation of *Ndr2* downregulated BV-2 cultured cells exposed, or not, to high-glucose conditions. WT BV-2 (WT) and *Ndr2* downregulated BV-2 (Clone 19 and Clone 22) cell cultures were incubated in 5.5 mM glucose (CT) or 30.5 mM glucose (HG) under different conditions: (**A**–**C**) 7 h assay; (**B**,**D**) 12 h assay. The results are presented as the mean ± SEM, and statistical analysis was assessed with a one-way ANOVA test, after confirmation of a Gaussian distribution; * *p* ≤ 0.05 compared with WT CT; *n* = 6–8 for Alamar Blue assay, and *n* = 5–7 for proliferation assay. EdU (green) incorporation indicates DNA synthesis during cell proliferation and cells were stained with the nuclear marker DAPI (blue).

**Figure 6 ijms-26-10630-f006:**
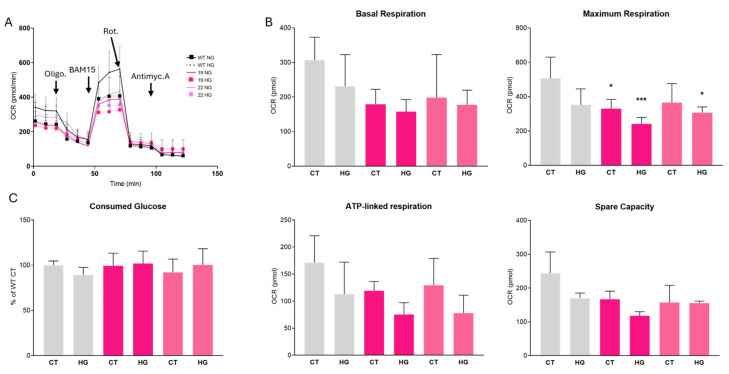
Metabolism of *Ndr2* downregulated BV-2 cells. Effect of *Ndr2* downregulation on mitochondrial metabolism in BV-2 microglial cells. (**A**,**B**) Mitochondrial oxygen consumption rate (OCR) measured using the Seahorse assay. OCR under NG conditions is represented by a solid line with circles, while HG conditions are represented by a dashed line with triangles. (**C**) Relative glucose consumption measured from culture media using a glucose test strip designed for diabetes monitoring. Measurements were taken after 7 h of exposure to NG or HG. Data are presented as mean ± SEM. Statistical analysis was performed using two-way ANOVA with Tukey’s post hoc correction. * *p* ≤ 0.05, *** *p* ≤ 0.001, compared to WT CT; *n* = 6–8.

**Figure 7 ijms-26-10630-f007:**
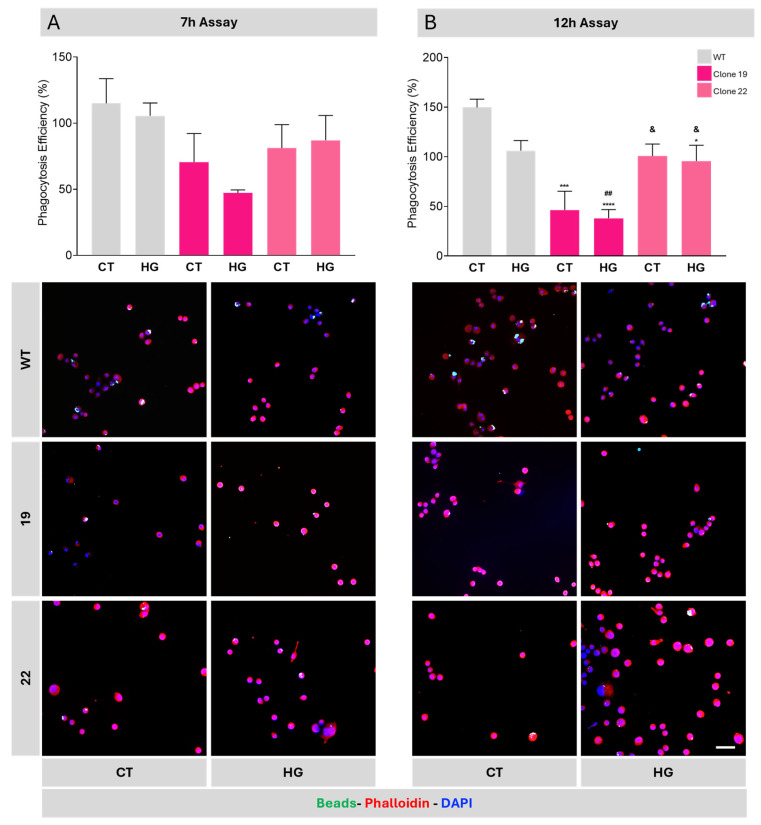
*Ndr2* downregulation decreases BV-2 phagocytic efficiency. WT and *Ndr2* downregulated BV-2 cell cultures were incubated in CT or HG conditions for the (**A**) 7 h assay; (**B**) 12 h assay. Fluorescent beads (green) were used to assess the phagocytic activity, and cells were stained with phalloidin (red) and the nuclear marker DAPI (blue). Scale bar: 50 µm. The results are presented as the mean ± SEM, and statistical analysis was assessed with a one-way ANOVA test after confirmation of a Gaussian distribution. * *p* ≤ 0.05, *** *p* ≤ 0.001, **** *p* ≤ 0.0001, compared with WT CT; ## *p* ≤ 0.001 compared with WT HG; & *p* ≤ 0.05, compared with clone 19 HG; *n* = 4–6.

**Figure 8 ijms-26-10630-f008:**
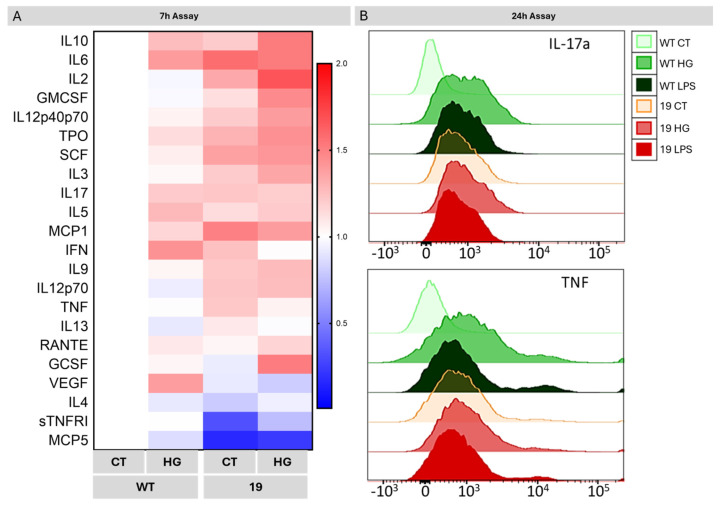
Expression of pro-inflammatory markers in WT and *Ndr2* downregulated BV-2 cells exposed to HG. (**A**) Cytokine array analysis demonstrates prominent upregulation of various pro-inflammatory factors secreted in the medium of *Ndr2* downregulated BV-2 cells following incubation in CT or HG for 7 h. (**B**) IL-17a and TNF levels were evaluated by flow cytometry analysis after 24 h incubation under CT, HG, or lipopolysaccharide (LPS; 100 ng/mL) stimulation. The images of the different membranes of the cytokine arrays and their map are represented in Appendix A.

**Table 1 ijms-26-10630-t001:** *Ndr2* Indel sequences. Three independent *Ndr2/Stk38l* alleles (Clone 13, Clone 19, and Clone 22), derived from separate clonal BV-2 populations, were analyzed by DNA sequencing. All clones harbor mutations in exon 7 of *Ndr2* (highlighted in red), resulting in frameshift mutations and reduced *Ndr2* expression. Results are expressed as fold change (FC) ± SEM (*n* = 2). The PAM sequence is shown in blue, the WT sequence in **black**, and the mutated sequence in red.

Clone	Indel Sequence	Length (bp)	FC	SEM
WT	**ACCAGACA**//TGCCAAGGCACGTGAGGAG GAGAGAGTCCTGGGCTTTGGCTGTGCCG**TGA**	60	0.49	
Clone 13	**ACCAGACA**//TGCCAAGGGACATGTAAAG TTATCCGATTTTGGTTTGTGCACGGGG**TGA**	59	0.49	0.04
Clone 19	**ACCAGACA**//TGCCAAGGGACATGTAAAA TTATCCGATTTTGGTTTGTGCACGGGG**TGA**	59	0.24	0.002
Clone 22	**ACCAGACA**//TGCCAAGGGACATGTAAAA GTTATCTGATTTTGGTTTGTGCACGGGGA**TGA**	61	0.27	0.02

**Table 2 ijms-26-10630-t002:** Oligonucleotide sequences. (A) single-guide RNA (sgRNA) sequence for against *Ndr2* exon 7 with the PAM sequence in **blue**; (B) Oligonucleotide primers for Sanger sequencing BV-2, targeting the *Ndr2* exons 7 and 8; (C) Oligonucleotide primers used for RT-qPCR, targeting the *Ywhaz* housekeeping gene and exons 13–14 of the mouse *Ndr1* and *Ndr2* genes. All the primers are reported with the symbol of the corresponding gene.

Name	Sequence
	A-Crispr plasmids
sgRNA #7	GCATCCAGTAAAAGGTTGTC**TGG**
	B-Sanger sequencing
*Ndr2 ex7 F*	5′-GTGACATGATGACATTGCTGATG-3′
*Ndr2 ex8 R*	5′-CCTCACACATAACCCGCCAAGC-3′
	C-RT-qPCR
*Ndr1 ex13 F*	5′-AAGCCCACAGTGACCACAAG-3′
*Ndr1 ex14 R*	5′-TGTACGAAGGTATGGCCCCC-3′
*Ndr2 ex13 F*	5′-GTTGAGAGGTCCATCCTGCC-3′
*Ndr3 ex14 R*	5′-CTGATTCTAGACCCACGGGC-3′
*Ywhaz F34*	5′-CAGCAAGCATACCAAGAAG-3′
*Ywhaz R35*	5′-TCGTAATAGAACACAGAGAAGT-3′

**Table 3 ijms-26-10630-t003:** Antibodies. List of antibodies used for immunocytochemistry (ICC), Western blot (WB) and flow cytometry (Flow cyt.). Antibodies are reported with the symbol of the corresponding protein (antigen), source including commercial company name, reference and concentrations used for either IHC or WB.

Primary Antibodies						
Antigen	Host	Company	Reference #	ICC conc.	WB conc.	Flow cyt.
Ndr1/2 (E-2)	Mouse monocl.	Santa Cruz Biotech, (Dallas, TX, USA)	sc-271703.	1/100	1/1000	-
Ndr2	Rabbit polycl.	St John’s lab. (London, UK)	STJ94368	1/100	1/1000	-
Iba1	Rabbit polycl.	Wako Chemicals USA (Richmond, VA, USA)	019-19741	1/500	-	-
GFAP	Chicken polycl.	Millipore (Burlington, MA, USA)	AV5541	1/100	-	-
Calnexin	Goat polycl.	Sicgen (Cantanhede, Portugal)	AB0041-500	-	1/1000	-
V450-A::IL-17a, clone N49-653	Mouse monocl.	BD Pharmingen (Franklin Lakes, NJ, USA)	RRID:AB_ 1727539	-	-	1x
PE-A::TNFa, clone TN3-19.12	Mouse monocl.	BioLegend (San Diego, CA, USA)	RRID:AB_315418	-	-	1x
CD11b-V500, clone M1/70	Mouse monocl.	BD Pharmingen (Franklin Lakes, NJ, USA)	RRID:AB_398535			
P2γ12-APC, clone S16607D	Mouse monocl.	BioLegend (San Diego, CA, USA)	RRID:AB_2721468			
FITC Annexin V Apoptosis Detection Kit with 7-AAD		BioLegend, San Diego, CA, USA	640922	-	-	1x
Phalloidin Tritc Labeled Mixed Isomers		Sigma-Aldrich (St Louis, MO, USA)	P1951	1x	-	-
**Secondary Antibodies**						
**Antigen**	**Host**	**Company**	**Reference #**	**IHC conc.**	**WB conc.**	**Fluorophore**
anti-Mouse IgG (H+L)	Goat polycl.	ThermoFisher Sc. (Waltham, MA, USA)	A11004	1/500	-	Alexa 568
anti-Rabbit IgG (H+L)	Goat polycl.	Invitrogen (Waltham, MA, USA)	A11008	1/500	-	Alexa 488
anti-Chicken IgG (H+L)	Goat polycl.	Invitrogen (Waltham, MA, USA)	A11041	1/500	-	Alexa 568
StarBright Blue 700 Goat Anti-Rabbit IgG	Goat polycl.	Bio-Rad (Hercules, CA, USA)	12004162	-	1/10,000	Blue 700
HRP Anti-Goat	Rabbit polycl.	Thermo Fisher Sc. (Waltham, MA, USA)	LTI 611620	-	1/10,000	-

## Data Availability

Data is contained within the article or Appendix A.

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
