# Peer review of "NDR2 Kinase Regulates Microglial Metabolic Adaptation and Inflammatory Response: Critical Role in Glucose-Dependent Functional Plasticity"

_ijms, 2025, doi:10.3390/ijms262110630_

Round 1
Reviewer 1 Report
Comments and Suggestions for Authors
The manuscript investigates NDR2 as a key regulator of microglial metabolism and inflammatory behavior under diabetic conditions. By modulating immune and metabolic responses, NDR2 may contribute to the neuroinflammatory processes underlying DR. Targeting NDR2 function in microglia may offer novel therapeutic strategies to mitigate retinal inflammation and progression of diabetic retinopathy.
While the study is well-written and addresses a relevant question, there are significant experimental design and interpretational limitations that need to be addressed before the conclusions can be fully supported:
1.Over-reliance on immortalized cell lines
Although primary mouse retinal microglia and human iPSC-derived microglia are described in the methods, the core functional assays appear to be performed exclusively in BV-2 cells. Given the phenotypic and transcriptional differences between BV-2 and primary microglia, key findings must be validated in primary retinal microglia and/or human iPSC-derived microglia to ensure physiological relevance.
2.Absence of in vivo confirmation
The proposed role of NDR2 in DR remains speculative without in vivo validation. A diabetic animal model (e.g., STZ-induced or db/db mice) with targeted microglial Ndr2 knockdown would provide critical confirmation that the observed effects occur in the retinal microenvironment.
3.High-glucose experimental design
The glucose concentration (30.5 mM) and exposure times (7–12 h) are short relative to chronic hyperglycemia in DR.
No osmotic control (e.g., mannitol) is included, making it impossible to distinguish hyperglycemia-specific effects from osmotic stress.
A time-course and dose–response analysis would strengthen the interpretation.
4.CRISPR/Cas9 editing validation
Only one knockdown strategy is used. No rescue (re-expression) or overexpression experiments are performed to confirm causality.
No off-target analysis (e.g., sequencing of top predicted sites) is provided, leaving uncertainty about specificity.
Clonal selection is described, but multiple independent clones are not compared, which risks clonal artifacts.
5.Mechanistic gap
The study reports phenotypic outcomes but does not measure downstream pathways linking NDR2 loss to mitochondrial dysfunction or cytoskeletal impairment (e.g., Hippo pathway activity, actin dynamics, mitochondrial morphology/function assays).
6.Human relevance not established
The discussion suggests clinical translation, but no data are presented on NDR2 expression in human DR retinal tissue or patient-derived microglia. This is necessary to support the therapeutic claims.
Author Response
Comments and Suggestions for Authors
The manuscript investigates NDR2 as a key regulator of microglial metabolism and inflammatory behavior under diabetic conditions. By modulating immune and metabolic responses, NDR2 may contribute to the neuroinflammatory processes underlying DR. Targeting NDR2 function in microglia may offer novel therapeutic strategies to mitigate retinal inflammation and progression of diabetic retinopathy.
While the study is well-written and addresses a relevant question, there are significant experimental design and interpretational limitations that need to be addressed before the conclusions can be fully supported:
- Over-reliance on immortalized cell lines
Although primary mouse retinal microglia and human iPSC-derived microglia are described in the methods, the core functional assays appear to be performed exclusively in BV-2 cells. Given the phenotypic and transcriptional differences between BV-2 and primary microglia, key findings must be validated in primary retinal microglia and/or human iPSC-derived microglia to ensure physiological relevance
We appreciate your comment and fully agree that phenotypic and transcriptional differences exist between BV-2 cells and primary/iPSC-derived microglia, with the latter being considered closer to in vivo physiology. However, the isolation of primary retinal microglia or the generation of iPSC-derived microglia is an extremely challenging, time-consuming process with low cellular yield, which limits their systematic use in extensive functional assays such as those performed in this study. Given these practical constraints, we opted to use the BV-2 cell line, a widely established and validated model for studying inflammatory mechanisms and microglial responses in vitro, which allowed us to ensure reproducibility and consistency in our results. Nevertheless, to increase the robustness of our study, we employed primary and iPSC-derived microglial cultures to characterize NDR2 localization—since this had never been demonstrated before—and to perform validation experiments, as detailed in the Methods section, thereby ensuring the biological relevance of the data obtained in BV-2 cells. We acknowledge that extensive validation in primary/iPSC-derived microglia would be ideal to further confirm the mechanisms observed. However, due to the technical and time limitations associated with their use, we believe that our experimental strategy represents an appropriate balance between practical feasibility and physiological relevance. This limitation has now been explicitly acknowledged and discussed in the Discussion section (Lines 521–532), as one of the limitations of the present study and as an important perspective for future research.
- Absence of in vivo confirmation
The proposed role of NDR2 in DR remains speculative without in vivo validation. A diabetic animal model (e.g., STZ-induced or db/db mice) with targeted microglial Ndr2 knockdown would provide critical confirmation that the observed effects occur in the retinal microenvironment.
We fully agree that in vivo validation is essential to strengthen the proposed role of NDR2 in DR. These studies using diabetic animal models with targeted microglial Ndr2 knockdown are currently ongoing in our laboratory, and the results will be reported in a separate manuscript, as the present work already contains a substantial amount of data.
- High-glucose experimental design
The glucose concentration (30.5 mM) and exposure times (7–12 h) are short relative to chronic hyperglycemia in DR. No osmotic control (e.g., mannitol) is included, making it impossible to distinguish hyperglycemia-specific effects from osmotic stress. A time-course and dose–response analysis would strengthen the interpretation.
We thank the reviewer for this valuable comment. The rationale behind the 7 h and 12 h exposures was not to mimic chronic hyperglycemia, but rather to reproduce the acute fluctuations in glucose levels that diabetic patients experience throughout the day. The glucose concentrations used (5.5 mM and 30.5 mM) were selected based on several previous studies conducted by our group, ensuring both consistency and comparability with our prior work.
Regarding osmotic control, experiments with mannitol were performed in multiple conditions by our group and others, and no significant differences were observed compared with the normal glucose control.
Indeed, several studies showed that high glucose exposure affects the inflammatory functions of macrophages, primary and immortalized microglial cells (Pavlou S, Lindsay J, Ingram R, Xu H, Chen M. Sustained high glucose exposure sensitizes macrophage responses to cytokine stimuli but reduces their phagocytic activity. BMC Immunol. 2018 Jul 11;19(1):24. doi: 10.1186/s12865-018-0261-0. PMID: 29996768; PMCID: PMC6042333; Lepiarz-Raba I, Gbadamosi I, Florea R, Paolicelli RC, Jawaid A. Metabolic regulation of microglial phagocytosis: Implications for Alzheimer's disease therapeutics. Transl Neurodegener. 2023 Oct 31;12(1):48. doi: 10.1186/s40035-023-00382-w. PMID: 37908010; PMCID: PMC10617244) while mannitol used as an osmotic control generally does not induce changes in microglial inflammatory markers or phagocytic functions (Xiang Zhang, Hongquan Dong, Susu Zhang, Shunmei Lu, Jie Sun, Yanning Qian; Enhancement of LPS-Induced Microglial Inflammation Response via TLR4 Under High Glucose Conditions. Cellular Physiology and Biochemistry 1 March 2015; 35 (4): 1571–1581; Wang Z, Lipshutz A, Martínez de la Torre C, Trzeciak AJ, Liu ZL, Miranda IC, Lazarov T, Codo AC, Romero-Pichardo JE, Nair A, Schild T, Saitz Rojas W, Saavedra PHV, Baako AK, Fadojutimi K, Downey MS, Geissmann F, Faraco G, Gan L, Etchegaray JI, Lucas CD, Tanasova M, Parkhurst CN, Zeng MY, Keshari KR, Perry JSA. Early life high fructose impairs microglial phagocytosis and neurodevelopment. Nature. 2025 Aug;644(8077):759-768. doi: 10.1038/s41586-025-09098-5. Epub 2025 Jun 11. PMID: 40500435; PMCID: PMC7617807).
However, we did perform mannitol controls of some of our experiments. Our first version contained a mannitol osmotic control of our resazurin assay which showed no difference of resazurin reduction demonstrating that the decrease of resazurin reduction and the decreased of mitochondrial metabolism of the Ndr2 downregulated BV-2 are not due to an osmotic effect but rather to the downregulation of Ndr2 expression. Our revised version contains also a phagocytosis assay perform with mannitol osmotic controls. The results show no difference of phagocytosis capability between the mannitol groups and their relative controls.
This information has now been included in the revised version of the manuscript (Supplementary Data, Lines 58–64 and Main Article, Lines 276-282).
- CRISPR/Cas9 editing validation
Only one knockdown strategy is used. No rescue (re-expression) or overexpression experiments are performed to confirm causality. No off-target analysis (e.g., sequencing of top predicted sites) is provided, leaving uncertainty about specificity. Clonal selection is described, but multiple independent clones are not compared, which risks clonal artifacts.
We thank the reviewer for highlighting this important point. While we did not perform experimental validation of potential off-targets, we have now conducted an in silico off-target analysis using the CRISPOR tool (http://crispor.tefor.net/), which predicts potential off-target sites for our sgRNA based on sequence similarity and genomic context. The sgRNA exhibited an efficiency score of 46.27 with the off-target tool of the CRISPR design website CHOPCHOP, coupled with limited off-target potential, showing only 2 and 1 predicted off-target transcripts with 1 and 3 mismatches, respectively. Further in silico analysis using CRISPOR confirmed its specificity with a MIT specificity score of 52/100, exceeding the recommended minimum threshold for guide RNA specificity, and a high CFD score of 91/100, indicating a strong likelihood of on-target cleavage with minimal off-target activity. Off-target predictions made with CRISPOR demonstrated that from the 15 first putative off-target, only 2 were intronic and 13 of them were intergenic, having a low likelihood of affecting our clones’phenotype. The analysis showed no predicted off-targets with high CFD scores or located in coding exons of genes related to inflammation, oxidative stress, or cytoskeletal regulation. This supports the specificity of our CRISPR/Cas9 targeting strategy. The results of the in silico analysis have been added to the revised Supplementary Table S2 (Supplementary Data, Lines 36–44) and described in the Methods section (Lines 641-643) and in results (Lines 149-165).
We agree that using multiple independent clones helps control for clonal variation. In our study, we analyzed two independently derived clones (Clone 19 and Clone 22), generated from the same sgRNA but carrying distinct indel mutations, as shown in Table 1. These clones exhibited consistent phenotypic and molecular changes, supporting that the observed effects are not due to clonal artifacts. We have clarified this point in the revised manuscript (Results section, Lines 190-194).
- Mechanistic gap
The study reports phenotypic outcomes but does not measure downstream pathways linking NDR2 loss to mitochondrial dysfunction or cytoskeletal impairment (e.g., Hippo pathway activity, actin dynamics, mitochondrial morphology/function assays).
We thank the reviewer for this valuable suggestion. We agree that investigating the downstream pathways linking NDR2 loss to mitochondrial dysfunction and cytoskeletal impairment would be an interesting avenue for future research. However, this is outside the scope of the current manuscript. We appreciate the reviewer's insight, which will certainly inform us about our future work on this topic. We have revised the discussion to acknowledge these limitations and suggest these pathways as potential areas for future studies (Discussion section, Lines 565-567).
- Human relevance not established
The discussion suggests clinical translation, but no data is presented on NDR2 expression in human diabetic retinopathy retinal tissue or patient-derived microglia. This is necessary to support the therapeutic claims.
We thank the reviewer for this valuable suggestion. We agree that direct evidence such as NDR2 expression in human DR retinal tissue or patient-derived microglia is essential to support translational claims. In this manuscript, we address this gap by demonstrating, via immunocytochemistry, that NDR2 kinase is indeed expressed in human iPSC-derived microglial cells. This provides the first direct evidence of NDR2 protein presence in a human microglial context relevant to retinal disease modeling. The use of iPSC-derived microglia is well-validated in literature as a physiologically relevant model, closely recapitulating the phenotype and function of primary human microglia. These cells have been shown to integrate into retinal tissue and adopt homeostatic and disease-relevant behaviors, further supporting the translational value of our findings. Additionally, the mammalian retina is highly conserved in terms of cellular composition and gene expression programs, with major retinal cell classes and many gene regulatory networks preserved across species. This conservation supports the relevance of findings from mammalian models to human retinal biology and pathology. Moreover, exploratory studies using human diabetic retinopathy retinal tissue and patient-derived microglia have recently been initiated in our laboratory. As these experiments are still at an early stage and fall outside the scope of the present manuscript, their results will be reported in a future publication.
In conclusion, we have thoroughly revised the manuscript to enhance English language accuracy and readability, corrected typographical errors, and implemented all suggested revisions. The funding section has also been updated to include the new grant information. We sincerely hope that these changes meet the reviewers’ expectations and contribute to a clearer and stronger manuscript.
Reviewer 2 Report
Comments and Suggestions for Authors
Summary:
The manuscript under review presents a well-designed and comprehensive in vitro study investigating the role of NDR2 kinase in microglial metabolic adaptation and inflammatory responses under high glucose (HG) conditions. The work is timely and relevant, given the central role of microglial activation and metabolic reprogramming in the pathogenesis of diabetic retinopathy (DR). The novelty lies in exploring the contribution of a relatively understudied Hippo pathway component, NDR2, to microglial function.
The study is generally well described, logically structured and integrates multiple methodologies, including protein and mRNA quantification, CRISPR/Cas9-mediated gene downregulation, Seahorse-based mitochondrial analysis, phagocytosis and migration assays, and cytokine profiling. The use of both 7-hour and 12-hour protocols provides valuable insight into acute hyperglycaemia and glucose fluctuation-related stress. The results are well-aligned with the stated aims and are supported by clear experimental design. The discussion effectively integrates current literature to contextualize the findings. The references are generally appropriate and up to date. The manuscript is written in clear and scholarly English and largely adheres to the standards of scientific writing.
Despite the aforementioned issues, the study offers important new insights into the role of NDR2 kinase in microglial metabolic and inflammatory regulation under hyperglycaemic conditions, identifying a potential therapeutic target for DR..Minor revisions are recommended to improve the clarity, consistency, and overall quality of the manuscript. Please find detailed comments below:
Comments:
- The authors acknowledge that complete knockout was not achieved, only partial downregulation. This important limitation should be explicitly stated in the Abstract to ensure transparency for readers.
- Some figure legends are dense, and it is occasionally unclear whether n refers to biological replicates or technical repeats. Clear specification is recommended.
- The manuscript alternates between “KO,” “knockdown,” and “downregulated cells.” A consistent terminology should be maintained throughout.
- Discussion Section could be expanded to address potential in vivo consequences of the observed cytokine profile changes, particularly regarding retinal vascular damage and neurodegeneration in DR.
- Line 46: “(American Academy of Ophthalmology 2025)” should be verified and replaced with a precise source (DOI or permanent URL).
- Recent literature (2022–2024) on microglial metabolic reprogramming could be added to further contextualise the findings.
- Line 116: In the phrase “primary retinal microglia was primarily localized…,” the repetition of “primary” and “primarily” is awkward. Rephrasing to avoid redundancy is recommended.
The manuscript is scientifically sound and represents a valuable contribution to the field. The suggested revisions are minor and aim to enhance the manuscript’s clarity, precision, and overall presentation.
Author Response
Comments and Suggestions for Authors
The manuscript under review presents a well-designed and comprehensive in vitro study investigating the role of NDR2 kinase in microglial metabolic adaptation and inflammatory responses under high glucose (HG) conditions. The work is timely and relevant, given the central role of microglial activation and metabolic reprogramming in the pathogenesis of diabetic retinopathy (DR). The novelty lies in exploring the contribution of a relatively understudied Hippo pathway component, NDR2, to microglial function.
The study is generally well described, logically structured and integrates multiple methodologies, including protein and mRNA quantification, CRISPR/Cas9-mediated gene downregulation, Seahorse-based mitochondrial analysis, phagocytosis and migration assays, and cytokine profiling. The use of both 7-hour and 12-hour protocols provides valuable insight into acute hyperglycaemia and glucose fluctuation-related stress. The results are well-aligned with the stated aims and are supported by clear experimental design. The discussion effectively integrates current literature to contextualize the findings. The references are generally appropriate and up to date. The manuscript is written in clear and scholarly English and largely adheres to the standards of scientific writing.
Despite the aforementioned issues, the study offers important new insights into the role of NDR2 kinase in microglial metabolic and inflammatory regulation under hyperglycaemic conditions, identifying a potential therapeutic target for DR. Minor revisions are recommended to improve the clarity, consistency, and overall quality of the manuscript.
The manuscript is scientifically sound and represents a valuable contribution to the field. The suggested revisions are minor and aim to enhance the manuscript’s clarity, precision, and overall presentation.
Please find detailed comments below:
Comments:
- The authors acknowledge that complete knockout was not achieved, only partial downregulation. This important limitation should be explicitly stated in the Abstract to ensure transparency for readers.
We thank the reviewer for this important observation. We agree that transparency regarding the level of gene disruption is crucial. We have revised the Abstract to clearly indicate that our CRISPR/Cas9 approach resulted in a partial downregulation of Ndr2, rather than a complete knockout (partially knocked out, Ndr2 knockdown, microglia with a partial downregulation of Ndr2). This change ensures that readers are appropriately informed of this limitation from the outset. The revised sentence now reads: “Using CRISPR-Cas9, we partially knocked out Ndr2/Stk38l gene in BV-2 mouse microglial cells …” (Abstract section, Lines 26-27)
- Some figure legends are dense, and it is occasionally unclear whether n refers to biological replicates or technical repeats. Clear specifications are recommended.
We thank the reviewer for pointing this out. To improve clarity and reproducibility, we have revised all figure legends to explicitly state whether n refers to biological replicates (independent experiments or cultures) or technical repeats (e.g., wells, images, or measurements from the same sample). The following sentence “Each n corresponds to data obtained from an independent culture in a separate experiment, except where otherwise stated.” was added to the material and methods section (Lines 610-612).
Additionally, we have simplified and reformatted dense figure legends which we hope offers improved readability. All changes are reflected in the revised manuscript.
- The manuscript alternates between “KO,” “knockdown,” and “downregulated cells.” A consistent terminology should be maintained throughout.
We appreciate the reviewer’s observation. To improve clarity and consistency, we have revised the manuscript to use the term “Ndr2 downregulated cells” when referring to our CRISPR/Cas9-edited microglial clones, which exhibit partial gene disruption rather than complete knockout. The term “KO” is now reserved exclusively for our planned BV-2 model and for models with confirmed complete gene deletion (e.g., primary microglia cells from Ndr2 KO mice). These adjustments have been applied consistently throughout the text and figure legends.
- Discussion Section could be expanded to address potential in vivo consequences of the observed cytokine profile changes, particularly regarding retinal vascular damage and neurodegeneration in DR.
We thank the reviewer for this insightful suggestion. In response, we have expanded the Discussion section to explore the potential in vivo implications of the altered cytokine profile observed in Ndr2-downregulated microglial cells. We discuss how this dysregulated inflammatory phenotype, characterized by elevated pro-inflammatory cytokines alongside altered anti-inflammatory signaling, may promote chronic neuroinflammation and vascular dysfunction. We highlight the role of microglia–endothelial crosstalk in driving endothelial activation, increased vascular permeability, and immune cell recruitment, processes increasingly recognized as early contributors to retinal inflammation and the pathogenesis of diabetic retinopathy. Our revised discussion places the findings in the context of microglial functional plasticity under metabolic stress and emphasizes NDR2’s central role in maintaining inflammatory homeostasis, with implications for neurodegenerative diseases including DR, Parkinson’s, and Alzheimer’s disease. This expanded section integrates recent insights into microglial metabolism, cytokine signaling, and vascular interactions that underlie retinal and CNS pathology in hyperglycemic conditions, addressing the reviewer’s suggestion comprehensively (Discussion, Lines 512–558).
- Line 46: “(American Academy of Ophthalmology 2025)” should be verified and replaced with a precise source (DOI or permanent URL).
We thank the reviewer for pointing this out. We have verified the source and replaced the citation with a precise and accessible reference (https://www.aao.org/) (Line 47). The references section was updated accordingly.
Recent literature (2022–2024) on microglial metabolic reprogramming could be added to further contextualize the findings.
We appreciate the reviewer’s suggestion and have incorporated recent literature (2022–2024) on microglial metabolic reprogramming to better contextualize our findings. These studies highlight ……. We have added this information to the revised Discussion section to strengthen the mechanistic interpretation of our results (Lines 507-542).
- Line 116: In the phrase “primary retinal microglia was primarily localized…,” the repetition of “primary” and “primarily” is awkward. Rephrasing to avoid redundancy is recommended.
We thank the reviewer for this stylistic suggestion. To improve clarity and readability, we have rephrased the sentence as follows: “In contrast, NDR2 staining in mouse primary retinal microglia was predominantly localized at the cell periphery and at the tips of microglial processes (Figure 2D).” (Page 03, Lines 113-115). This change removes the redundancy while preserving the intended meaning. The correction has been implemented in the revised manuscript.
In conclusion, we have thoroughly revised the manuscript to enhance English language accuracy and readability, corrected typographical errors, and implemented all suggested revisions. The funding section has also been updated to include the new grant information. We sincerely hope that these changes meet the reviewers’ expectations and contribute to a clearer and stronger manuscript.
Round 2
Reviewer 1 Report
Comments and Suggestions for Authors
The author answered all my questions.